# DeepLTL: Learning to Efficiently Satisfy Complex LTL Specifications for Multi-Task RL

**Mathias Jackermeier, Alessandro Abate**
Department of Computer Science, University of Oxford
{mathias.jackermeier,alessandro.abate}@cs.ox.ac.uk

## Abstract

Linear temporal logic (LTL) has recently been adopted as a powerful formalism for specifying complex, temporally extended tasks in multi-task reinforcement learning (RL). However, learning policies that efficiently satisfy arbitrary specifications not observed during training remains a challenging problem. Existing approaches suffer from several shortcomings: they are often only applicable to finite-horizon fragments of LTL, are restricted to suboptimal solutions, and do not adequately handle safety constraints. In this work, we propose a novel learning approach to address these concerns. Our method leverages the structure of Büchi automata, which explicitly represent the semantics of LTL specifications, to learn policies conditioned on sequences of truth assignments that lead to satisfying the desired formulae. Experiments in a variety of discrete and continuous domains demonstrate that our approach is able to zero-shot satisfy a wide range of finite- and infinite-horizon specifications, and outperforms existing methods in terms of both satisfaction probability and efficiency. Code available at: https://deep-ltl.github.io/

## 1 Introduction

One of the fundamental challenges in artificial intelligence (AI) is to create agents capable of following arbitrary instructions. While significant research efforts have been devoted to designing reinforcement learning (RL) agents that can complete tasks expressed in natural language (Oh et al., 2017; Goyal et al., 2019; Luketina et al., 2019), recent years have witnessed increased interest in *formal* languages to specify tasks in RL (Andreas et al., 2017; Camacho et al., 2019; Jothimurugan et al., 2021). Formal specification languages offer several desirable properties over natural language, such as well-defined semantics and compositionality, allowing for the specification of unambiguous, structured tasks (Vaezipoor et al., 2021; León et al., 2022). Recent works have furthermore shown that it is possible to automatically translate many natural language instructions to formal languages, providing interpretable yet precise representations of tasks (Pan et al., 2023; Liu et al., 2023).

*Linear temporal logic* (LTL) (Pnueli, 1977) in particular has been adopted as a powerful formalism for instructing RL agents (Hasanbeig et al., 2018; Araki et al., 2021; Voloshin et al., 2023). LTL is an appealing specification language that allows for the definition of tasks in terms of high-level features of the environment. These tasks can utilise complex compositional structure, are inherently temporally extended (i.e. non-Markovian), and naturally incorporate safety constraints.

While several approaches have been proposed to learning policies capable of zero-shot executing arbitrary LTL specifications in a multi-task RL setting (Kuo et al., 2020; Vaezipoor et al., 2021; Qiu et al., 2023; Liu et al., 2024), they suffer from several limitations. First, most existing methods are limited to subsets of LTL and cannot handle infinite-horizon (i.e. $\omega$-regular) specifications, which form an important class of objectives including *persistence* (eventually, a desired state needs to be reached forever), *recurrence* (a set of states needs to be reached infinitely often), and *response* (whenever a particular event happens, a task needs to be completed) (Manna & Pnueli, 1990). Second, many current techniques are *myopic*, that is, they solve tasks by independently completing individual subtasks, which can lead to inefficient, globally suboptimal solutions (Vaezipoor et al., 2021). Finally, existing approaches often do not adequately handle safety constraints of specifications, especially when tasks can be completed in multiple ways with different safety implications. For an illustration of these limitations, see Figure 1.

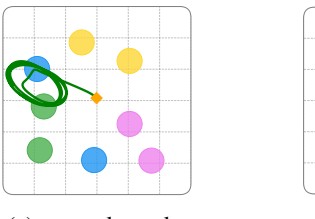 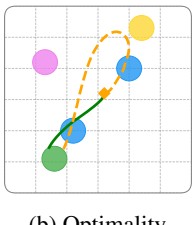 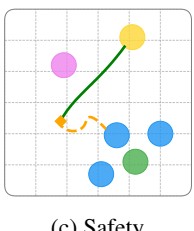

(a) $\omega$-regular tasks                (b) Optimality                (c) Safety

Figure 1: Limitations of existing methods, illustrated via trajectories in the *ZoneEnv* environment. The initial agent position is denoted as an orange diamond. (a) Most existing approaches cannot handle infinite-horizon tasks, such as $\mathsf{G}\,\mathsf{F}$ blue $\wedge\,\mathsf{G}\,\mathsf{F}$ green. (b) Given the formula $\mathsf{F}$ (blue $\wedge\,\mathsf{F}$ green), a *myopic* approach produces a suboptimal solution (orange line). We prefer the more efficient green trajectory. (c) Given the task ($\mathsf{F}$ green $\vee\,\mathsf{F}$ yellow) $\wedge\,\mathsf{G}\,\neg$blue, the agent should aim to reach the yellow region instead of the green region, since this is the safer option. Many existing approaches are unable to perform this sort of planning.

In this paper, we develop a novel approach to learning policies for zero-shot execution of LTL specifications that addresses these shortcomings. Our method exploits the structure of Büchi automata to non-myopically reason about ways of completing a (possibly infinite-horizon) specification, and to ensure that safety constraints are satisfied. Our main contributions are as follows:

- we develop the *first* non-myopic approach to learning multi-task policies for zero-shot execution of LTL specifications that is applicable to infinite-horizon tasks;
- we propose a novel representation of LTL formulae based on reach-avoid sequences of truth assignments, which allows us to learn policies that intrinsically consider safety constraints;
- we propose a novel neural network architecture that combines DeepSets and RNNs to condition the policy on the desired specification;
- lastly, we empirically validate the effectiveness of our method on a range of environments and tasks, demonstrating that it outperforms existing approaches in terms of satisfaction probability and efficiency.

## 2  BACKGROUND

**Reinforcement learning.**    We model RL environments using the framework of *Markov decision processes* (MDPs). An MDP is a tuple $\mathcal{M} = (\mathcal{S}, \mathcal{A}, \mathcal{P}, \mu, r, \gamma)$, where $\mathcal{S}$ is the state space, $\mathcal{A}$ is the set of actions, $\mathcal{P}\colon \mathcal{S} \times \mathcal{A} \to \Delta(\mathcal{S})$ is the *unknown* transition kernel, $\mu \in \Delta(\mathcal{S})$ is the initial state distribution, $r\colon \mathcal{S} \times \mathcal{A} \times \mathcal{S} \to \mathbb{R}$ is the reward function, and $\gamma \in [0, 1)$ is the discount factor.

We denote the probability of transitioning from state $s$ to state $s'$ after taking action $a$ as $\mathcal{P}(s' \mid s, a)$. A (memoryless) *policy* $\pi\colon \mathcal{S} \to \Delta(\mathcal{A})$ is a map from states to probability distributions over actions. Executing a policy $\pi$ in an MDP gives rise to a trajectory $\tau = (s_0, a_0, r_0, s_1, a_1, r_1, \dots)$, where $s_0 \sim \mu$, $a_t \sim \pi(\cdot \mid s_t)$, $s_{t+1} \sim \mathcal{P}(\cdot \mid s_t, a_t)$, and $r_t = r(s_t, a_t, s_{t+1})$. The goal of RL is to find a policy $\pi^*$ that maximises the *expected discounted return* $J(\pi) = \mathbb{E}_{\tau \sim \pi}\left[\sum_{t=0}^{\infty} \gamma^t r_t\right]$, where we write $\tau \sim \pi$ to indicate that the distribution over trajectories depends on the policy $\pi$. The *value function* of a policy $V^\pi(s) = \mathbb{E}_{\tau \sim \pi}\left[\sum_{t=0}^{\infty} \gamma^t r_t \mid s_0 = s\right]$ is defined as the expected discounted return starting from state $s$ and following policy $\pi$ thereafter.

**Linear temporal logic.**    Linear temporal logic (LTL) (Pnueli, 1977) provides a formalism to precisely specify properties of infinite trajectories. LTL formulae are defined over a set of atomic propositions $AP$, which describe high-level features of the environment. The syntax of LTL formulae is recursively defined as

$$\mathsf{true} \mid a \mid \varphi \wedge \psi \mid \neg\varphi \mid \mathsf{X}\,\varphi \mid \varphi\,\mathsf{U}\,\psi$$

where $a \in AP$ and $\varphi$ and $\psi$ are themselves LTL formulae. $\wedge$ and $\neg$ are the Boolean operators conjunction and negation, which allow for the definition of all standard logical connectives. The temporal operators $\mathsf{X}$ and $\mathsf{U}$ intuitively mean "next" and "until". We define the two temporal modalities $\mathsf{F}$ ("eventually") and $\mathsf{G}$ ("always") as $\mathsf{F}\,\varphi \equiv \mathsf{true}\,\mathsf{U}\,\varphi$ and $\mathsf{G}\,\varphi \equiv \neg\mathsf{F}\,\neg\varphi$.

The semantics of LTL align with the intuitive meanings of its operators. For example, in the *ZoneEnv* environment depicted in Figure 1, the atomic propositions $AP$ correspond to coloured regions. We can naturally express many desirable tasks as LTL specifications, such as reaching a blue region (F blue), avoiding blue until a yellow region is reached (¬blue U yellow), reaching and remaining in a green region (F G green), or oscillating between blue and green regions while avoiding yellow (G F green ∧ G F blue ∧ G ¬yellow). The latter two examples represent *infinite-horizon* specifications, which describe behaviour over an infinite time horizon.

Formally, the satisfaction semantics of LTL are defined via a recursive satisfaction relation $w \models \varphi$ on infinite sequences $w$ of truth assignments[1] over $AP$ (i.e. $\omega$-words over $2^{AP}$) (see Appendix A for details). To ground LTL specifications in an MDP, we assume access to a *labelling function* $L \colon \mathcal{S} \to 2^{AP}$, which returns the atomic propositions that are true in a given state. A trajectory $\tau$ is mapped to a sequence of assignments via its trace $\mathrm{Tr}(\tau) = L(s_0)L(s_1)\dots$, and we write $\tau \models \varphi$ as shorthand for $\mathrm{Tr}(\tau) \models \varphi$. The *satisfaction probability* of an LTL formula $\varphi$ under policy $\pi$ is then defined as $\Pr(\pi \models \varphi) = \mathbb{E}_{\tau \sim \pi}\big[\mathbb{1}[\tau \models \varphi]\big]$.

**Büchi automata.** A more practical way of reasoning about the semantics of LTL formulae is via *Büchi automata* (Büchi, 1966), which are specialised automata that can be constructed to keep track of the progress towards satisfying a specification. In particular, in this work we focus on *limit-deterministic Büchi automata* (LDBAs) (Sickert et al., 2016), which are particularly amenable to be employed with MDPs. An LDBA is a tuple $\mathcal{B} = (\mathcal{Q}, q_0, \Sigma, \delta, \mathcal{F}, \mathcal{E})$, where $\mathcal{Q}$ is a finite set of states, $q_0 \in \mathcal{Q}$ is the initial state, $\Sigma = 2^{AP}$ is a finite alphabet, $\delta \colon \mathcal{Q} \times (\Sigma \cup \mathcal{E}) \to \mathcal{Q}$ is the transition function, and $\mathcal{F}$ is the set of accepting states. Additionally, $\mathcal{Q}$ is composed of two disjoint subsets $\mathcal{Q} = \mathcal{Q}_N \uplus \mathcal{Q}_D$ such that $\mathcal{F} \subseteq \mathcal{Q}_D$ and $\delta(q, \alpha) \in \mathcal{Q}_D$ for all $q \in \mathcal{Q}_D$ and $\alpha \in \Sigma$. The set $\mathcal{E}$ is an indexed set of $\varepsilon$-transitions (a.k.a jump transitions), which transition from $\mathcal{Q}_N$ to $\mathcal{Q}_D$ without consuming any input, and there are no other transitions from $\mathcal{Q}_N$ to $\mathcal{Q}_D$.

A *run* $r$ of $\mathcal{B}$ on the $\omega$-word $w$ is an infinite sequence of states in $\mathcal{Q}$ respecting the transition function. An infinite word $w$ is *accepted* by $\mathcal{B}$ if there exists a run $r$ of $\mathcal{B}$ on $w$ that visits an accepting state infinitely often. For every LTL formula $\varphi$, we can construct an LDBA $\mathcal{B}_\varphi$ that accepts exactly the words satisfying $\varphi$ (Sickert et al., 2016).

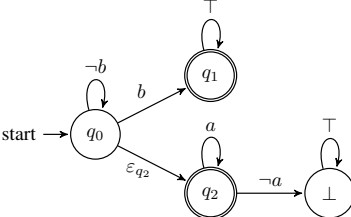

**Example 1.** Figure 2 depicts an LDBA for the formula (F G a) ∨ F b. The automaton starts in state $q_0$ and transitions to the accepting state $q_1$ upon observing the proposition $b$. Once it has reached $q_1$, it stays there indefinitely. Alternatively, it can

Figure 2: LDBA for the formula (F G a) ∨ F b.

transition to the accepting state $q_2$ without consuming any input via the $\varepsilon$-transition. Once in $q_2$, the automaton accepts exactly the words where $a$ is true at every step. ☐

## 3 PROBLEM SETTING

Our high-level goal is to find a specification-conditioned policy $\pi | \varphi$ that maximises the probability of satisfying arbitrary LTL formulae $\varphi$. Formally, we introduce an arbitrary distribution $\xi$ over LTL specifications $\varphi$, and aim to compute the optimal policy

$$\pi^* = \arg\max_{\pi} \mathbb{E}_{\substack{\varphi \sim \xi, \\ \tau \sim \pi | \varphi}} \big[\mathbb{1}[\tau \models \varphi]\big]. \tag{1}$$

We now introduce the necessary formalism to find solutions to Equation 1 via reinforcement learning.

**Definition 1** (Product MDP). The *product MDP* $\mathcal{M}^\varphi$ of an MDP $\mathcal{M}$ and an LDBA $\mathcal{B}_\varphi$ synchronises the execution of $\mathcal{M}$ and $\mathcal{B}_\varphi$. Formally, $\mathcal{M}^\varphi$ has the state space $\mathcal{S}^\varphi = \mathcal{S} \times \mathcal{Q}$, action space $\mathcal{A}^\varphi = \mathcal{A} \times \mathcal{E}$, initial state distribution $\mu^\varphi(s, q) = \mu(s) \cdot \mathbb{1}[q = q_0]$, and transition function

$$\mathcal{P}^\varphi\left((s', q') \mid (s, q), a\right) = \begin{cases} \mathcal{P}(s' \mid s, a) & \text{if } a \in \mathcal{A} \wedge q' = \delta(q, L(s)), \\ 1 & \text{if } a = \varepsilon_{q'} \wedge q' = \delta(q, a) \wedge s' = s, \\ 0 & \text{otherwise.} \end{cases}$$

---

[1]An *assignment* $a$ is a subset of $AP$. Propositions $p \in a$ are assigned *true*, whereas $p \notin a$ are assigned *false*.

The product MDP $\mathcal{M}^\varphi$ extends the state space of $\mathcal{M}$ in order to keep track of the current state of the LDBA. This allows us to consider only *memoryless* policies that map tuples $(s, q)$ of MDP and LDBA states to actions, since the LDBA takes care of adding the memory necessary to satisfy $\varphi$ (Baier & Katoen, 2008). Quite importantly, note that in practice we never build the product MDP explicitly, but instead simply update the current LDBA state $q$ with the propositions $L(s)$ observed at each time step. Also note that the action space in $\mathcal{M}^\varphi$ is extended with $\mathcal{E}$ to allow the policy to follow $\varepsilon$-transitions in $\mathcal{B}_\varphi$ without acting in the MDP. Trajectories in $\mathcal{M}^\varphi$ are sequences $\tau = ((s_0, q_0), a_0, (s_1, q_1), a_1, \dots)$, and we denote as $\tau_q$ the projection of $\tau$ onto the LDBA states $q_0, q_1, \dots$. We can restate the satisfaction probability of formula $\varphi$ in $\mathcal{M}^\varphi$ as

$$\Pr(\pi \models \varphi) = \mathop{\mathbb{E}}_{\tau \sim \pi} \left[ \mathbb{1} \left[ \inf(\tau_q) \cap \mathcal{F} \neq \emptyset \right] \right],$$

where $\inf(\tau_q)$ denotes the set of states that occur infinitely often in $\tau_q$.

An optimal policy can be found via goal-conditioned reinforcement learning in the product MDP as follows: given some probability distribution $\xi$ over LTL formulae, sample $\varphi \sim \xi$ and trajectories $\tau \sim \pi|\varphi$, assigning a positive reward of 1 whenever the policy visits an accepting state in the LDBA corresponding to $\varphi$, and 0 otherwise. To ensure that the reward-maximising policy $\pi_{RL}^*$ indeed maximises Equation 1, one can employ *eventual discounting* (Voloshin et al., 2023) and only discount visits to accepting states in the automaton and not the steps between those visits (for a further discussion, see Appendix B.1). With this, we obtain the following bound on the performance of $\pi_{RL}^*$:

**Theorem 1** (Corollary of Voloshin et al. (2023), Theorem 4.2). *For any $\gamma \in (0, 1)$ we have*

$$\sup_\pi \mathop{\mathbb{E}}_{\varphi \sim \xi} \left[ \Pr(\pi \models \varphi) \right] - \mathop{\mathbb{E}}_{\varphi \sim \xi} \left[ \Pr(\pi_{RL}^* \models \varphi) \right] \leq 2 \log(\frac{1}{\gamma}) \sup_\pi O_\pi,$$

*where $O_\pi = \mathbb{E}_{\varphi \sim \xi, \tau \sim \pi|\varphi} \left[ |\{ q \in \tau_q : q \in \mathcal{F}_{\mathcal{B}_\varphi} \}| \, \big| \, \tau \not\models \varphi \right]$ is the expected number of visits to accepting states for trajectories that do not satisfy a specification.*

*Proof.* The proof follows from (Voloshin et al., 2023, Theorem 4.2) by repeated application of the linearity of expectation and triangle inequality. A detailed proof is given in Appendix B.2. □

However, while eventual discounting provides desirable theoretical guarantees, it does not take into account any notion of *efficiency* of formula satisfaction, which is an important practical concern. Consider for example the simple formula F a. Eventual discounting assigns the same return to all policies that eventually visit $s$ with a $\in L(s)$, regardless of the number of steps required to materialise a. In practice, we often prefer policies that are more efficient (require fewer steps to make progress towards satisfaction), even if their overall satisfaction probability might be slightly reduced. A natural way to formalise this tradeoff is as follows:

**Problem 1** (Efficient LTL satisfaction). Given an unknown MDP $\mathcal{M}$, a distribution over LTL formulae $\xi$, and LDBAs $\mathcal{B}_\varphi$ for each $\varphi \in \text{supp}(\xi)$ (the support of $\xi$), find the optimal policy

$$\pi^* = \arg\max_\pi \mathop{\mathbb{E}}_{\substack{\varphi \sim \xi, \\ \tau \sim \pi|\varphi}} \left[ \sum_{t=0}^\infty \gamma^t \mathbb{1}[q_t \in \mathcal{F}_{\mathcal{B}_\varphi}] \right]. \tag{2}$$

Here, we discount *all* time steps, such that more efficient policies receive higher returns. While solutions to Problem 1 are not guaranteed to be probability-optimal (as per Equation 1), they will generally still achieve a high probability of formula satisfaction, while also taking efficiency into account. We consider Problem 1 for the rest of this paper, since we believe efficiency to be an important practical concern, but note that our approach is equally applicable to the eventual discounting setting (see Appendix B.3). There is an interesting connection to *discounted LTL* (Almagor et al., 2014; Alur et al., 2023) (see Appendix C for details), but we leave a full investigation thereof for future work.

## 4 METHOD

Solving Problem 1 requires us to train a policy conditioned on the current MDP state $s$ and the current state $q$ of an LDBA constructed from a given target specification $\varphi$. Our key insight is that we can extract a useful representation of $q$ directly from the structure of the LDBA, by reasoning about the possible ways of satisfying the given formula from the current LDBA state $q$. This representation is then used to condition the policy, and guide the agent towards satisfying a given specification.

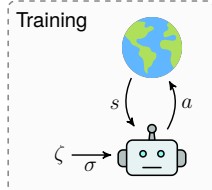 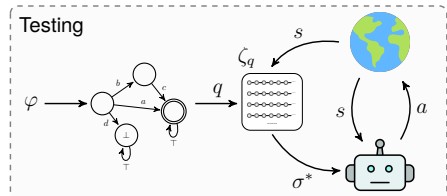

Figure 3: Overview of our approach. (*Left*) During training, we train a general sequence-conditioned policy with reach-avoid sequences $\sigma$. (*Right*) At test time, we construct an LDBA from the target specification $\varphi$. We then select the optimal reach-avoid sequence $\sigma^*$ for the current LDBA state $q$ according to the value function $V^\pi(s, \sigma)$, and produce an action $a$ from the policy conditioned on $\sigma^*$.

## 4.1 REPRESENTING LTL SPECIFICATIONS AS SEQUENCES

**Computing accepting cycles.** An optimal policy for Problem 1 must continuously visit accepting states in $\mathcal{B}_\varphi$. Since $\mathcal{B}_\varphi$ is finite, this means that the agent has to reach an *accepting cycle* in the LDBA. Intuitively, the possible ways of reaching accepting cycles are an informative representation of the current LDBA state $q$, as they capture how to satisfy the given task. We compute all possible ways of reaching an accepting cycle using an algorithm based on depth-first search (DFS) that explores all possible paths from $q$ to an accepting state $q_f \in \mathcal{F}$, and then back to a state already contained in the path (see Appendix E for details). Let $P_q$ denote the resulting set of paths from $q$ to accepting cycles.

*Remark.* In the case that $\varphi$ corresponds to a task that can be completed in finite time (e.g. F a), the accepting cycle in $\mathcal{B}_\varphi$ is trivial and consists of only a single looping state (see e.g. $q_1$ in Figure 2). We note that our algorithm to compute accepting cycles is similar to previous work on reward shaping (Shah et al., 2024); however, we use the accepting cycles for an altogether different purpose.

**From paths to sequences.** A path $p \in P_q$ is an infinite sequence of states $(q_1, q_2, \ldots)$ in the LDBA. Let $A_i^+ = \{a : \delta(q_i, a) = q_{i+1}\}$ denote the set of assignments $a \in 2^{AP}$ that progress the LDBA from state $q_i$ to $q_{i+1}$.[2] We furthermore define the set of negative assignments $A_i^- = \{a : a \notin A_i^+ \wedge \delta(q_i, a) \neq q_i\}$ that lead from $q_i$ to a different state in the LDBA (excluding self-loops). In order to satisfy the LTL specification via $p$, the policy has to sequentially visit MDP states $s_t$ such that $L(s_{t_i}) \in A_i^+$ for some $t_i$, while avoiding assignments in $A_i^-$. We refer to the sequence

$$\sigma_p = \big((A_1^+, A_1^-), (A_2^+, A_2^-), \ldots\big)$$

as the *reach-avoid sequence* corresponding to $p$, and denote as $\zeta_q = \{\sigma_p : p \in P_q\}$ the set of all reach-avoid sequences for $q$.

**Example 2.** The first two steps of $\sigma = \big((\{\{a\}\}, \{\{b, d\}\}), (\{\{c\}, \{e\}\}, \emptyset), \ldots\big)$ require the agent to achieve a while avoiding states with both b and d, and subsequently achieve c or e. □

## 4.2 OVERVIEW OF THE APPROACH

See Figure 3 for an overview of our method. Representing the current LDBA state $q$ as a set of reach-avoid sequences allows us to condition the policy on possible ways of achieving the given specification. On a high level, our approach works as follows: in the *training stage*, we learn a general sequence-conditioned policy $\pi : \mathcal{S} \times \zeta \to \Delta(\mathcal{A})$ together with its value function $V^\pi : \mathcal{S} \times \zeta \to \mathbb{R}$ to satisfy arbitrary reach-avoid sequences $\sigma \in \zeta$ over $AP$, following the standard framework of *goal-conditioned* RL (Liu et al., 2022). Note that we do not assume access to a distribution $\xi$ over formulae, since we are interested in satisfying arbitrary specifications. At *test time*, we are now given a target specification $\varphi$ and construct its corresponding LDBA. We keep track of the current LDBA state $q$, and select the optimal reach-avoid sequence to follow in order to satisfy $\varphi$ according to the value function of $\pi$, i.e.

$$\sigma^* = \arg\max_{\sigma \in \zeta_q} V^\pi(s, \sigma). \tag{3}$$

We then execute actions $a \sim \pi(\cdot, \sigma^*)$ until the next LDBA state is reached.

---

[2]For now, we assume that there are no $\varepsilon$-transitions in $p$. We revisit $\varepsilon$-transitions in Section 4.5.

$$\sigma = \left((A_1^+, A_1^-), (A_2^+, A_2^-), \dots\right) \xrightarrow{\textit{DeepSets}} \left(\boldsymbol{e}_{A_1^+}\|\boldsymbol{e}_{A_1^-},\ \boldsymbol{e}_{A_2^+}\|\boldsymbol{e}_{A_2^-}, \dots\right) \xrightarrow{\textit{RNN}} \boldsymbol{e}_{\sigma}$$

Figure 4: Illustration of the *sequence module*. The positive and negative assignments in the truncated reach-avoid sequence $\sigma$ are encoded using the *DeepSets* architecture, which produces embeddings $\boldsymbol{e}_A$. These are then concatenated and passed through an RNN, which yields the final sequence representation $\boldsymbol{e}_{\sigma}$.

The test-time execution of our approach can be equivalently thought of as executing a policy $\widetilde{\pi}$ in the product MDP $\mathcal{M}^{\varphi}$, where $\widetilde{\pi}(s, q) = \pi(s, \sigma^*)$. That is, $\widetilde{\pi}$ exploits $\pi$ to reach an accepting cycle in the LDBA of the target specification, and thus approximates Problem 1. Next, we describe the model architecture of the sequence-conditioned policy, and give a detailed description of the training procedure and test-time execution.

## 4.3 MODEL ARCHITECTURE

We parameterise the sequence-conditioned policy $\pi$ using a modular neural network architecture. This consists of an *observation module*, which processes observations from the environment, a *sequence module*, which encodes the reach-avoid sequence, and an *actor module*, which takes as input the features produced by the previous two modules and outputs a distribution over actions.

The *observation module* is implemented as either a fully-connected (for generic state features) or convolutional neural network (for image-like observations). The *actor module* is another fully connected neural network that outputs the mean and standard deviation of a Gaussian distribution (for continuous action spaces) or the parameters of a categorical distribution (in the discrete setting). Finally, the *sequence module* consists of a permutation-invariant neural network that encodes sets of assignments, and a recurrent neural network (RNN) that maps the resulting sequence to a final representation. We discuss these components in further detail below and provide an illustration of the sequence module in Figure 4.

**Representing sets of assignments.** The first step of the sequence module consists in encoding the sets of assignments in a reach-avoid sequence. We employ the *DeepSets* architecture (Zaheer et al., 2017) to obtain an encoding $\boldsymbol{e}_A$ of a set of assignments $A$. That is, we have

$$\boldsymbol{e}_A = \rho\left(\sum_{a \in A} \phi(a)\right), \tag{4}$$

where $\phi(a)$ is a learned embedding function, and $\rho$ is a learned non-linear transformation. Note that the resulting encoding $\boldsymbol{e}_A$ is *permutation-invariant*, i.e. it does not depend on the order in which the elements in $A$ are processed, and Equation 4 is thus a well-defined function on sets.

**Representing reach-avoid sequences.** Once we have obtained encodings of the sets $A_i^+$ and $A_i^-$ for each element in the reach-avoid sequence $\sigma$, we concatenate these embeddings and pass them through an RNN to yield the final representation of the sequence. Since $\sigma$ is an infinite sequence, we approximate it with a finite prefix by repeating its looping part $k$ times, such that the truncated sequence visits an accepting state exactly $k$ times. We apply the RNN backwards, that is, from the end of the truncated sequence to the beginning, since earlier elements in $\sigma$ are more important for the immediate actions of the policy. The particular model of RNN we employ is a *gated recurrent unit* (GRU) (Cho et al., 2014).

## 4.4 TRAINING PROCEDURE

We train the policy $\pi$ and the value function $V^{\pi}$ using the general framework of *goal-conditioned RL* (Liu et al., 2022). That is, we generate a random reach-avoid sequence at the beginning of each training episode and reward the agent for successfully completing it. In particular, given a training sequence $\sigma = \left((A_1^+, A_1^-), \dots, (A_n^+, A_n^-)\right)$, we keep track of the task satisfaction progress via an index $i \in [n]$ (where initially $i = 1$). We say the agent *satisfies* a set of assignments $A$ at time step $t$ if $L(s_t) \in A$. Whenever the agent satisfies $A_i^+$, we increment $i$ by one. If $i = n + 1$, we

give the agent a reward of $1$ and terminate the episode. If the agent at any point satisfies $A_i^-$, we also terminate the episode and give it a negative reward of $-1$. Otherwise, the agent receives zero reward. By maximising the expected discounted return, the policy learns to efficiently satisfy arbitrary reach-avoid sequences. In our experiments, we use *proximal policy optimisation* (PPO) (Schulman et al., 2017) to optimise the policy, but our approach can be combined with any RL algorithm.

**Curriculum learning.** To improve the training of $\pi$ in practice, we employ a simple form of *curriculum learning* (Narvekar et al., 2020) in order to gradually expose the policy to more challenging tasks. A curriculum consists of multiple stages that correspond to training sequences of increasing length and complexity. For example, the first stage generally consists only of simple reach-tasks of the form $\sigma = \big( (\{\{p\}\}, \emptyset) \big)$ for $p \in AP$, while later stages involve longer sequences with avoidance conditions. Once the policy achieves satisfactory performance on the current tasks, we move on to the next stage. For details on the exact curricula we use in our experiments, see Appendix F.4.

## 4.5 TEST TIME POLICY EXECUTION

At test time, we execute the trained sequence-conditioned policy $\pi$ to complete an arbitrary task $\varphi$. As described in Section 4.2, we keep track of the current LDBA state $q$ in $\mathcal{B}_\varphi$, and use the learned value function $V^\pi$ to select the optimal reach-avoid sequence $\sigma^*$ to follow from $q$ in order to satisfy $\varphi$ (Equation 3). Note that it follows from the reward of our training procedure that

$$V^\pi(s, \sigma) \leq \mathop{\mathbb{E}}_{\tau \sim \pi | \sigma} \left[ \sum_{t=0}^{\infty} \gamma^t \mathbb{1}[i = n+1] \,\middle|\, s_0 = s \right],$$

i.e. the value function is a lower bound of the discounted probability of reaching an accepting state $k$ times via $\sigma$ (where $k$ is the number of loops in the truncated sequence). As $k \to \infty$, the sequence $\sigma^*$ that maximises $V^\pi$ thus maximises a lower bound on Problem 1 for the trained policy $\pi$. Once $\sigma^*$ has been selected, we execute actions $a \sim \pi(\cdot, \sigma^*)$ until the next LDBA state is reached.

**Strict negative assignments.** Recall that a negative assignment in a reach-avoid sequence $\sigma_p$ is any assignment that leads to an LDBA state other than the desired next state in $p$. In practice, we find that trying to avoid all other states in the LDBA can be too restrictive for the policy. We therefore only regard as negative those assignments that lead to a significant reduction in expected performance. In particular, given a threshold $\lambda$, we define the set of *strict* negative assignments for state $q_i \in p$ as the assignments that lead to a state $q'$ where

$$V^\pi(s, \sigma_p[i \dots]) - \max_{\sigma' \in \zeta_{q'}} V^\pi(s, \sigma') \geq \lambda.$$

We then set $A_i^-$ to be the set of *strict* negative assignments for $q_i$. Reducing $\lambda$ leads to a policy that more closely follows the selected path $p$, whereas increasing $\lambda$ gives the policy more flexibility to deviate from the chosen path.

**Handling $\varepsilon$-transitions.** We now discuss how to handle $\varepsilon$-transitions in the LDBA. As described in Section 2, whenever the LDBA is in a state $q$ with an $\varepsilon$-transition to $q'$, the policy can choose to either stay in $q$ or transition to $q'$ without acting in the MDP. If the sequence $\sigma^*$ chosen at $q$ starts with an $\varepsilon$-transition (i.e. $A_1^+ = \{\varepsilon\}$), we extend the action space of $\pi$ to include the action $\varepsilon$. If $\mathcal{A}$ is discrete, we simply add an additional dimension to the action space. In the continuous case, we learn the probability $p$ of taking the $\varepsilon$-action and model $\pi(\cdot | s, \sigma^*)$ as a mixed continuous/discrete probability distribution (see e.g. (Shynk, 2012, Ch. 3.6)). Whenever the policy executes the $\varepsilon$-action, we update the current LDBA state to the next state in the selected path. In practice, we additionally only allow $\varepsilon$-actions if $L(s) \notin A_2^-$, since in that case taking the $\varepsilon$-transition would immediately lead to failure.

## 4.6 DISCUSSION

We argue that our approach has several advantages over existing methods. Since we operate on accepting cycles of Büchi automata, our method is applicable to infinite-horizon (i.e. $\omega$-regular) tasks, contrary to most existing approaches. Our method is the *first* approach that is also non-myopic, as it is able to reason about the entire structure of a specification via temporally extended reach-avoid

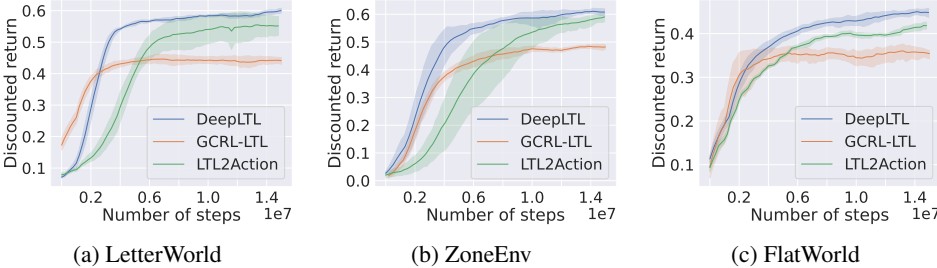

Figure 5: Evaluation curves on *reach/avoid* specifications. Each datapoint is collected by averaging the discounted return of the policy across 50 episodes with randomly sampled tasks, and shaded areas indicate 90% confidence intervals over 5 different random seeds.

sequences. This reasoning naturally considers safety constraints, which are represented through negative assignments and inform the policy about which propositions to avoid. Crucially, these safety constraints are considered during planning, i.e. when selecting the optimal sequence to execute, rather than only during execution. For a detailed comparison of our approach to related work, see Section 6.

## 5 EXPERIMENTS

We evaluate our approach, called *DeepLTL*, in a variety of environments and on a range of LTL specifications of varying difficulty. We aim to answer the following questions: **(1)** Is DeepLTL able to learn policies that can zero-shot satisfy complex LTL specifications? **(2)** How does our method compare to relevant baselines in terms of both satisfaction probability and efficiency? **(3)** Can our approach successfully handle infinite-horizon specifications?

### 5.1 EXPERIMENTAL SETUP

**Environments.** Our experiments involve different domains with varying state and action spaces. This includes the *LetterWorld* environment (Vaezipoor et al., 2021), a $7 \times 7$ discrete grid world in which letters corresponding to atomic propositions occupy randomly sampled positions in the grid. We also consider the high-dimensional *ZoneEnv* environment from Vaezipoor et al. (2021), in which a robotic agent with a continuous action space has to navigate between different randomly placed coloured regions, which correspond to the atomic propositions. Finally, we evaluate our approach on the continuous *FlatWorld* environment (Voloshin et al., 2023), in which multiple propositions can hold true at the same time. We provide further details and visualisations in Appendix F.1.

**LTL specifications.** We consider a range of tasks of varying complexity. *Reach/avoid* specifications are randomly sampled from a task space that encompasses both sequential reachability objectives of the form $\mathsf{F}\,(p_1 \wedge (\mathsf{F}\,p_2 \wedge (\mathsf{F}\,p_3)))$ and reach-avoid tasks $\neg p_1 \,\mathsf{U}\,(p_2 \wedge (\neg p_3 \,\mathsf{U}\,p_4))$, where the $p_i$ are randomly sampled atomic propositions. *Complex* specifications are given by more complicated, environment-specific LTL formulae, such as the specification $((\mathsf{a} \vee \mathsf{b} \vee \mathsf{c} \vee \mathsf{d}) \Rightarrow \mathsf{F}\,(\mathsf{e} \wedge (\mathsf{F}\,(\mathsf{f} \wedge \mathsf{F}\,\mathsf{g})))) \,\mathsf{U}\,(\mathsf{h} \wedge \mathsf{F}\,\mathsf{i})$ in *LetterWorld*. We also separately investigate *infinite-horizon* tasks such as $\mathsf{G}\,\mathsf{F}\,\mathsf{a} \wedge \mathsf{G}\,\mathsf{F}\,\mathsf{b}$ and $\mathsf{F}\,\mathsf{G}\,\mathsf{a}$. The specifications we consider cover a wide range of LTL objectives, including reachability, safety, recurrence, persistence, and combinations thereof. Details on the exact specifications we use in each environment are given in Appendix F.2.

**Baselines.** We compare DeepLTL to two state-of-the-art approaches for learning general LTL-satisfying policies. LTL2Action (Vaezipoor et al., 2021) encodes the syntax tree of a target formula via a graph neural network (GNN) and uses a procedure known as *LTL progression* to progress through the specification based on the observed propositions. The second baseline, GCRL-LTL (Qiu et al., 2023), instead learns proposition-conditioned policies and combines them compositionally using a weighted graph search on the Büchi automaton of a target specification.

**Evaluation protocol.** In line with previous work, the methods are trained for 15M interaction steps on each environment with PPO (Schulman et al., 2017). Details about hyperparameters and

Table 1: Evaluation results of trained policies on *complex* finite-horizon specifications. We report the *success rate* (SR) and average number of steps to satisfy the task ($\mu$). Results are averaged over 5 seeds and 500 episodes per seed. "$\pm$" indicates the standard deviation over seeds.

| | | LTL2Action | | GCRL-LTL | | DeepLTL | |
|---|---|---|---|---|---|---|---|
| | | SR ($\uparrow$) | $\mu$ ($\downarrow$) | SR ($\uparrow$) | $\mu$ ($\downarrow$) | SR ($\uparrow$) | $\mu$ ($\downarrow$) |
| LetterWorld | $\varphi_1$ | $0.75_{\pm 0.18}$ | $29.48_{\pm 3.20}$ | $0.94_{\pm 0.05}$ | $15.29_{\pm 0.70}$ | $\mathbf{1.00}_{\pm 0.00}$ | $\mathbf{9.66}_{\pm 0.35}$ |
| | $\varphi_2$ | $0.79_{\pm 0.10}$ | $19.04_{\pm 6.79}$ | $0.94_{\pm 0.03}$ | $9.77_{\pm 1.16}$ | $\mathbf{0.98}_{\pm 0.00}$ | $\mathbf{7.26}_{\pm 0.35}$ |
| | $\varphi_3$ | $0.41_{\pm 0.14}$ | $40.31_{\pm 2.88}$ | $\mathbf{1.00}_{\pm 0.00}$ | $20.72_{\pm 1.34}$ | $\mathbf{1.00}_{\pm 0.00}$ | $12.23_{\pm 0.58}$ |
| | $\varphi_4$ | $0.72_{\pm 0.17}$ | $28.83_{\pm 4.47}$ | $0.82_{\pm 0.07}$ | $14.60_{\pm 1.63}$ | $\mathbf{0.97}_{\pm 0.01}$ | $12.13_{\pm 0.58}$ |
| | $\varphi_5$ | $0.44_{\pm 0.26}$ | $31.84_{\pm 9.06}$ | $\mathbf{1.00}_{\pm 0.00}$ | $25.63_{\pm 0.55}$ | $\mathbf{1.00}_{\pm 0.00}$ | $\mathbf{9.48}_{\pm 0.78}$ |
| ZoneEnv | $\varphi_6$ | $0.60_{\pm 0.20}$ | $424.07_{\pm 14.95}$ | $0.85_{\pm 0.03}$ | $452.19_{\pm 15.59}$ | $\mathbf{0.92}_{\pm 0.06}$ | $\mathbf{303.38}_{\pm 19.43}$ |
| | $\varphi_7$ | $0.14_{\pm 0.18}$ | $416.78_{\pm 66.38}$ | $0.85_{\pm 0.05}$ | $451.18_{\pm 04.91}$ | $\mathbf{0.91}_{\pm 0.03}$ | $\mathbf{299.95}_{\pm 09.47}$ |
| | $\varphi_8$ | $0.67_{\pm 0.26}$ | $414.48_{\pm 68.52}$ | $0.89_{\pm 0.04}$ | $449.70_{\pm 16.82}$ | $\mathbf{0.96}_{\pm 0.04}$ | $\mathbf{259.75}_{\pm 08.07}$ |
| | $\varphi_9$ | $0.69_{\pm 0.22}$ | $331.55_{\pm 41.40}$ | $0.87_{\pm 0.02}$ | $303.13_{\pm 05.83}$ | $\mathbf{0.90}_{\pm 0.03}$ | $\mathbf{203.36}_{\pm 14.97}$ |
| | $\varphi_{10}$ | $0.66_{\pm 0.19}$ | $293.22_{\pm 63.94}$ | $0.85_{\pm 0.02}$ | $290.73_{\pm 17.39}$ | $\mathbf{0.91}_{\pm 0.02}$ | $\mathbf{187.13}_{\pm 10.61}$ |
| | $\varphi_{11}$ | $0.93_{\pm 0.07}$ | $123.89_{\pm 07.30}$ | $0.89_{\pm 0.01}$ | $137.42_{\pm 08.30}$ | $\mathbf{0.98}_{\pm 0.01}$ | $\mathbf{106.21}_{\pm 07.88}$ |
| FlatWorld | $\varphi_{12}$ | $\mathbf{1.00}_{\pm 0.00}$ | $83.32_{\pm 01.57}$ | $0.82_{\pm 0.41}$ | $\mathbf{78.21}_{\pm 08.98}$ | $\mathbf{1.00}_{\pm 0.00}$ | $79.69_{\pm 02.50}$ |
| | $\varphi_{13}$ | $0.63_{\pm 0.50}$ | $94.43_{\pm 39.30}$ | $0.00_{\pm 0.00}$ | $0.00_{\pm 0.00}$ | $\mathbf{1.00}_{\pm 0.00}$ | $52.82_{\pm 03.09}$ |
| | $\varphi_{14}$ | $0.71_{\pm 0.40}$ | $96.16_{\pm 28.93}$ | $0.73_{\pm 0.41}$ | $74.60_{\pm 01.86}$ | $\mathbf{0.98}_{\pm 0.01}$ | $71.76_{\pm 02.87}$ |
| | $\varphi_{15}$ | $0.07_{\pm 0.02}$ | $\mathbf{32.37}_{\pm 01.63}$ | $0.73_{\pm 0.03}$ | $41.30_{\pm 01.24}$ | $\mathbf{0.86}_{\pm 0.01}$ | $43.87_{\pm 01.45}$ |
| | $\varphi_{16}$ | $0.56_{\pm 0.35}$ | $48.85_{\pm 32.85}$ | $0.64_{\pm 0.08}$ | $\mathbf{17.76}_{\pm 01.63}$ | $\mathbf{1.00}_{\pm 0.01}$ | $37.04_{\pm 05.28}$ |

neural network architectures can be found in Appendix F.3. We report the performance in terms of discounted return over the number of environment interactions (following Equation 2) on randomly sampled *reach/avoid* tasks, and provide tabular results detailing the success rate (SR) and average number of steps until completion ($\mu$) of trained policies on *complex* tasks. All results are averaged across 5 different random seeds. Furthermore, we provide visualisations of trajectories of trained policies for various specifications in the *ZoneEnv* and *FlatWorld* environments in Appendix G.9.

## 5.2 RESULTS

**Finite-horizon tasks.** Figure 5 shows the discounted return achieved on *reach/avoid* tasks across environment interactions. DeepLTL clearly outperforms the baselines, both in terms of sample efficiency and final performance. The results in Table 1 further demonstrate that our method can efficiently zero-shot satisfy *complex* specifications (see Appendix F.2 for details on the tasks), achieving higher success rates (SR) than existing approaches, while requiring significantly fewer steps ($\mu$). These results highlight the performance benefits of our representation based on reach-avoid sequences over existing encoding schemes, and show that our approach learns much more efficient policies than the myopic baseline GCRL-LTL. The higher success rates of our method furthermore indicate that it handles safety constraints better than the baselines.

**Infinite-horizon tasks.** Figure 6 shows example trajectories of DeepLTL on infinite-horizon tasks in the *ZoneEnv* environment. In Figure 6c we furthermore compare the performance of our approach on recurrence, i.e. G F , tasks (see Appendix F.2 for details) to GCRL-LTL, the only previous approach that can handle infinite-horizon specifications. We report the average number of visits to accepting states per episode, which corresponds to the number of completed cycles of target propositions (e.g. the number of times both blue and green have been visited for the specification G F blue $\land$ G F green). Additional results on F G tasks can be found in Appendix G.1. Our evaluation confirms that DeepLTL can successfully handle $\omega$-regular tasks, and significantly outperforms the only relevant baseline.

**Further experimental results.** We provide further experimental results in Appendix G. These include more results on tasks with safety requirements (G.2), an investigation of the generalisation capabilities of DeepLTL to longer sequences (G.3), two ablation studies on the impact of curriculum learning (G.4) and our model architecture (G.5), investigations on the performance impact of the number of atomic propositions (G.6) and specification complexity (G.7), and a comparison to *RAD-embeddings* (Yalcinkaya et al., 2024), a recent automata-conditioned RL method (G.8). Finally, we show trajectories of trained policies in the *ZoneEnv* and *FlatWorld* environments in Appendix G.9.

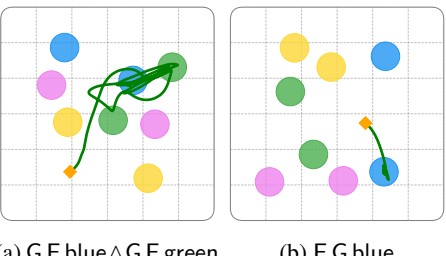

| | | GCRL-LTL | DeepLTL |
|---|---|---|---|
| LetterWorld | $\psi_1$ | $6.74_{\pm 1.96}$ | $\mathbf{17.84}_{\pm 0.67}$ |
| | $\psi_2$ | $1.49_{\pm 0.31}$ | $\mathbf{3.05}_{\pm 0.47}$ |
| ZoneEnv | $\psi_3$ | $3.14_{\pm 0.18}$ | $\mathbf{4.16}_{\pm 0.61}$ |
| | $\psi_4$ | $1.31_{\pm 0.05}$ | $\mathbf{1.71}_{\pm 0.23}$ |
| FlatWorld | $\psi_5$ | $7.20_{\pm 4.08}$ | $\mathbf{12.12}_{\pm 0.80}$ |
| | $\psi_6$ | $4.64_{\pm 0.25}$ | $\mathbf{6.98}_{\pm 1.00}$ |

(a) G F blue∧G F green     (b) F G blue         (c)

Figure 6: Results on infinite-horizon tasks. (a), (b) Example trajectories for infinite-horizon specifications. (c) Performance on various recurrence tasks. We report the average number of visits to accepting states over 500 episodes (i.e. completed cycles), with standard deviations over 5 seeds.

## 6 RELATED WORK

RL with tasks expressed in LTL has received significant attention in the last few years (Sadigh et al., 2014; De Giacomo et al., 2018; Camacho et al., 2019; Kazemi et al., 2022; Li et al., 2024). Our approach builds on previous works that use LDBAs to augment the state space of the MDP (Hasanbeig et al., 2018; Hahn et al., 2019; 2020; Bozkurt et al., 2020; Voloshin et al., 2022; Hasanbeig et al., 2023; Bagatella et al., 2024; Shah et al., 2024). However, these methods are limited to finding policies for a *single*, fixed specification. In contrast, our approach is realised in a multi-task setting and learns a policy that can zero-shot generalise to arbitrary specifications at test time.

Among the works that consider multiple, previously unseen specifications, many approaches decompose a given task into subtasks, which are then individually completed (Araki et al., 2021; León et al., 2021; 2022; Liu et al., 2024). However, as noted by Vaezipoor et al. (2021) this results in *myopic* behaviour and hence potentially suboptimal solutions. In contrast, our approach takes the entire specification into account by reasoning over temporally extended reach-avoid sequences. Kuo et al. (2020) instead propose to compose RNNs in a way that mirrors formula structure, which however requires learning a non-stationary policy. This is addressed by LTL2Action (Vaezipoor et al., 2021), which encodes the syntax tree of a target specification using a GNN and uses *LTL progression* (Bacchus & Kabanza, 2000) to make the problem Markovian. A similar approach is proposed by Yalcinkaya et al. (2024), who use a GNN to encode tasks represented by deterministic finite automata. We instead extract reach-avoid sequences from Büchi automata, which explicitly lists the possible ways of satisfying a given specification. Furthermore, due to its reliance on LTL progression, LTL2Action is restricted to the finite-horizon fragment of LTL, whereas our approach is able to handle infinite-horizon tasks.

The only previous method we are aware of that can deal with infinite-horizon specifications is GCRL-LTL (Qiu et al., 2023). However, similar to other approaches, GCRL-LTL relies on composing policies for sub-tasks and therefore produces suboptimal behaviour. Furthermore, the approach only considers safety constraints during task execution and not during high-level planning. Recently, Xu & Fekri (2024) proposed *future dependent options* for satisfying arbitrary LTL tasks, which are option policies that depend on future goals. Their method is only applicable to a fragment of LTL that does not support conjunction nor infinite-horizon specifications, and does not consider safety constraints during planning. See Appendix D for an extended discussion of related work.

## 7 CONCLUSION

We have introduced *DeepLTL*, a novel approach to the problem of learning policies that can zero-shot execute arbitrary LTL specifications. Our method represents a given specification as a set of reach-avoid sequences of truth assignments, and exploits a general sequence-conditioned policy to execute arbitrary LTL instructions at test time. In contrast to existing techniques, our method can handle infinite-horizon specifications, is non-myopic, and naturally considers safety constraints. Through extensive experiments, we have demonstrated the effectiveness of our approach in practice. This work contributes to the field of LTL-conditioned RL, which has the potential to pave the way towards general AI systems capable of executing arbitrary, well-defined tasks.

ACKNOWLEDGEMENTS

We are grateful for the valuable feedback from the anonymous reviewers. This work was supported by the UKRI AI Hub (EP/Y028872/1) and the ARIA TA1 project SAINT. MJ is funded by the EPSRC Centre for Doctoral Training in Autonomous Intelligent Machines and Systems (EP/S024050/1).

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

## A    LTL SATISFACTION SEMANTICS

The satisfaction semantics of LTL are defined in terms of infinite sequences of truth assignments $a \in 2^{AP}$ (a.k.a. $\omega$-words over $2^{AP}$). The satisfaction relation $w \models \varphi$ specifies that $\omega$-word $w$ *satisfies* the specification $\varphi$. It is recursively defined as follows (Baier & Katoen, 2008):

$$
\begin{aligned}
w &\models \mathsf{true} \\
w &\models \mathsf{a} &&\text{iff } \mathsf{a} \in w_0 \\
w &\models \varphi \wedge \psi &&\text{iff } w \models \varphi \text{ and } w \models \psi \\
w &\models \neg\varphi &&\text{iff } w \not\models \varphi \\
w &\models \mathsf{X}\,\varphi &&\text{iff } w[1\ldots] \models \varphi \\
w &\models \varphi \,\mathsf{U}\, \psi &&\text{iff } \exists j \geq 0 \text{ s.t. } w[j\ldots] \models \psi \text{ and } \forall 0 \leq i < j. \; w[i\ldots] \models \varphi.
\end{aligned}
$$

As noted in the main paper, we can equivalently define the satisfaction semantics via (limit-deterministic) Büchi automata. Formally, for any LTL specification $\varphi$ we can construct a Büchi automaton that accepts exactly the set $\mathrm{Words}(\varphi) = \{w \in (2^{AP})^\omega \mid w \models \varphi\}$.

## B    LEARNING PROBABILITY-OPTIMAL POLICIES

### B.1    EVENTUAL DISCOUNTING

The technique of *eventual discounting* (Voloshin et al., 2023) can be used to find an optimal policy for a given (single) LTL specification $\varphi$ in terms of satisfaction probability via reinforcement learning. It optimises the following objective:

$$
\pi^*_{RL} = \arg\max_\pi \; \mathbb{E}_{\tau \sim \pi}\left[\sum_{t=0}^\infty \Gamma_t \mathbb{1}[q_t \in \mathcal{F}_{\mathcal{B}_\varphi}]\right], \quad \Gamma_t = \gamma^{c_t}, \quad c_t = \sum_{k=0}^t \mathbb{1}[q_k \in \mathcal{F}_{\mathcal{B}_\varphi}],
$$

where $c_t$ counts how often accepting states have been visited up to time step $t$.

To see why eventual discounting is necessary, consider the *product* MDP $\mathcal{M}^\varphi$ depicted in Figure 7, and adapted from Voloshin et al. (2023). The policy starts in state $s_0$ and can choose either action $a$ or action $b$. Action $a$ always leads to an infinite cycle containing an accepting state, and is thus optimal. Action $b$ on the other hand also leads to an infinite cycle with probability 0.99, but may lead to a sink state with probability 0.01.

Let $\pi_a$ be the (deterministic) policy that chooses $a$ and $\pi_b$ be the policy that chooses $b$. Without eventual discounting, we have:

$$
J(\pi_a) = \frac{1}{1-\gamma^2} \qquad \text{and} \qquad J(\pi_b) = \frac{0.99}{1-\gamma},
$$

and thus $J(\pi_b) > J(\pi_a)$ for all $\gamma \in (0.01, 1)$. Hence, maximising (standard) expected discounted return produces a suboptimal policy in terms of satisfaction probability.

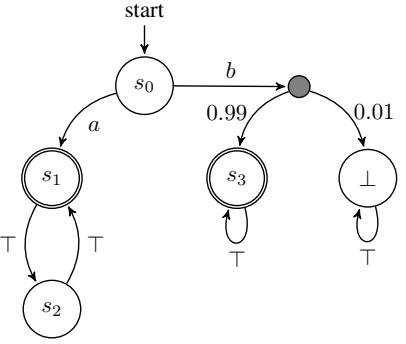

Figure 7: Example product MDP.

Eventual discounting addresses this problem by only discounting visits to accepting states, and not the steps in between. In the previous example, this means that $J(\pi_a) > J(\pi_b)$, in line with the satisfaction probability. See Voloshin et al. (2023) for a formal derivation of a bound on the performance of the return-optimal policy under eventual discounting. We next extend this result to the setting of a distribution $\xi$ over LTL specifications $\varphi$.

### B.2  PROOF OF THEOREM 1

Given a probability distribution $\xi$ over LTL specifications $\varphi$, we employ the standard formulation of goal-conditioned RL to obtain the multi-task eventual discounting objective:

$$\pi_{RL}^* = \arg\max_\pi \mathop{\mathbb{E}}_{\substack{\varphi \sim \xi, \\ \tau \sim \pi|\varphi}} \left[ \sum_{t=0}^\infty \Gamma_t \mathbb{1}[q_t \in \mathcal{F}_{\mathcal{B}_\varphi}] \right], \quad \Gamma_t = \gamma^{c_t}, \quad c_t = \sum_{k=0}^t \mathbb{1}[q_k \in \mathcal{F}_{\mathcal{B}_\varphi}]. \tag{5}$$

Theorem 1 provides a bound on the performance of the policy $\pi_{RL}^*$. Our proof closely follows the structure of the proof of (Voloshin et al., 2023, Theorem 4.2). We begin with the following Lemma:

**Lemma 1.** *For any $\pi$, $\gamma \in (0, 1)$, and $\varphi \in \mathrm{supp}(\xi)$, we have*

$$|(1 - \gamma)V^\pi - \Pr(\pi \models \varphi)| \leq \log(\frac{1}{\gamma})O_\pi,$$

*where $O_\pi = \mathbb{E}_{\varphi \sim \xi, \tau \sim \pi|\varphi}\big[|\{q \in \tau_q : q \in \mathcal{F}_{\mathcal{B}_\varphi}\}|\,\big|\,\tau \not\models \varphi\big]$ is the expected number of visits to accepting states for trajectories that do not satisfy a specification.*

*Proof.* The proof follows exactly along the lines of the proof of (Voloshin et al., 2023, Lemma 4.1) with our modified definition of $O_\pi$, which includes the expectation over $\varphi$. $\square$

We are now ready to prove the main result:

**Theorem 1.** *For any $\gamma \in (0, 1)$ we have*

$$\sup_\pi \mathop{\mathbb{E}}_{\varphi \sim \xi}[\Pr(\pi \models \varphi)] - \mathop{\mathbb{E}}_{\varphi \sim \xi}[\Pr(\pi_{RL}^* \models \varphi)] \leq 2\log(\frac{1}{\gamma})\sup_\pi O_\pi.$$

*Proof.* Let $(\pi_i)_{i \in \mathbb{N}}$ be a sequence of policies such that

$$\mathop{\mathbb{E}}_{\varphi \sim \xi}[\Pr(\pi_i \models \varphi)] \xrightarrow{i \to \infty} \sup_\pi \mathop{\mathbb{E}}_{\varphi \sim \xi}[\Pr(\pi \models \varphi)].$$

By the linearity of expectation, we have

$$\mathop{\mathbb{E}}_{\varphi \sim \xi}[\Pr(\pi_i \models \varphi)] - \mathop{\mathbb{E}}_{\varphi \sim \xi}[\Pr(\pi_{RL}^* \models \varphi)]$$
$$= \mathop{\mathbb{E}}_{\varphi \sim \xi}[\Pr(\pi_i \models \varphi) - \Pr(\pi_{RL}^* \models \varphi)].$$

We add and subtract the terms $(1 - \gamma)V^{\pi_i}$ and $(1 - \gamma)V^{\pi_{RL}^*}$ and apply the triangle inequality to obtain

$$\leq \left| \mathop{\mathbb{E}}_\varphi[\Pr(\pi_i \models \varphi) - (1 - \gamma)V^{\pi_i}] \right| + \left| \mathop{\mathbb{E}}_\varphi\left[\Pr(\pi_{RL}^* \models \varphi) - (1 - \gamma)V^{\pi_{RL}^*}\right] \right|$$
$$+ \mathop{\mathbb{E}}_\varphi\left[(1 - \gamma)V^{\pi_i} - (1 - \gamma)V^{\pi_{RL}^*}\right]. \tag{6}$$

The last term is negative since

$$\mathop{\mathbb{E}}_\varphi\left[(1 - \gamma)V^{\pi_i} - (1 - \gamma)V^{\pi_{RL}^*}\right] = (1 - \gamma)\mathop{\mathbb{E}}_\varphi\left[V^{\pi_i} - V^{\pi_{RL}^*}\right]$$
$$= (1 - \gamma)(V^{\pi_i} - V^{\pi_{RL}^*})$$
$$\leq 0 \quad \text{(by the definition of } \pi_{RL}^*\text{)}.$$

Note that for any random variable $X$, again by the triangle inequality,

$$|\mathbb{E}[X]| = \left| \sum_x x \Pr(X = x) \right| \leq \sum_x |x| \Pr(X = x) = \mathbb{E}[|X|],$$

and we can hence continue from Equation 6 by applying Lemma 1 as follows (where we utilise the fact the expectations respect inequalities):

$$(6) \leq \mathbb{E}_\varphi \left[ \log(\frac{1}{\gamma})(O_{\pi_i} + O_{\pi^*_{RL}}) \right]$$

$$\leq \mathbb{E}_\varphi \left[ 2\log(\frac{1}{\gamma}) \sup_\pi O_\pi \right]$$

$$= 2\log(\frac{1}{\gamma}) \sup_\pi O_\pi,$$

which, together with taking the limit as $i \to \infty$, concludes the proof. $\qquad\square$

### B.3 DEEPLTL WITH EVENTUAL DISCOUNTING

DeepLTL can be readily extended with eventual discounting for settings in which satisfaction probability is the primary concern, and efficiency is less important. In this case, we want to use our approach to approximate a solution to Equation 5.

To do so, we only need to assume access to the distribution $\xi$ over LTL formulae. During training, we sample specifications $\varphi \sim \xi$ and train the sequence-conditioned policy using all reach-avoid sequences extracted from $\mathcal{B}_\varphi$. Crucially, we extend each step of a reach-avoid sequence to include an additional Boolean flag that specifies whether the corresponding LDBA transition leads to an accepting state. These flags are given as input to the policy network, and are used to compute the eventual discounting objective. The rest of our approach remains unchanged. We leave an experimental investigation of this scheme for future work.

## C DISCOUNTED LTL

*Discounted LTL* (Almagor et al., 2014) extends LTL with discounting semantics. This allows one to specify not only that a system should satisfy a given specification (as in classical LTL), but also *how well* it should do so. The main idea is to replace the "until" operator with a discounted version that assigns a quantitative satisfaction value in $[0, 1]$ to a system trace. Intuitively, the longer the system takes to satisfy a requirement, the smaller the assigned satisfaction value.

The problem of finding an optimal policy with respect to a discounted LTL formula in an unknown MDP has been thoroughly investigated by Alur et al. (2023), and is closely related to our problem of efficient LTL satisfaction (Problem 1). Indeed, for simple formulae such as $\varphi = \mathsf{F}_\gamma\, a$ (where $\mathsf{F}_\gamma$ denotes the $\gamma$-discounted version of the $\mathsf{F}$ operator) it is easy to see that Equation 2 corresponds exactly to maximising the quantitative satisfaction value of $\varphi$. We leave a fuller investigation of the relationship between discounting rewards from accepting LDBA states and the semantics of discounted LTL, as well as an integration of our approach with the reward scheme proposed by Alur et al. (2023), for future work.

## D EXTENDED RELATED WORK

The field of RL with LTL specifications has attracted significant attention in the last few years. Here we provide a more detailed overview of work in this domain and discuss how it relates to our approach.

**RL with a single LTL specification.** Early works on RL with LTL specifications relied on estimating a model of the underlying MDP, and then solving this model for a probability-maximising policy (Fu & Topcu, 2014; Brázdil et al., 2014; Sadigh et al., 2014). A model-free approach based on $Q$-learning (Watkins, 1989; Watkins & Dayan, 1992) was introduced by Hasanbeig et al. (2018), who

proposed to use LDBAs to keep track of formula satisfaction. Subsequent works extend this approach, providing stronger convergence guarantees (Hahn et al., 2019; Bozkurt et al., 2020; Hahn et al., 2023; Voloshin et al., 2023), improved sample efficiency (Hahn et al., 2020; Kazemi & Soudjani, 2020; Cai et al., 2021; Shao & Kwiatkowska, 2023; Shah et al., 2024; Bagatella et al., 2024), or guarantees in adversarial environments (Bozkurt et al., 2024). Alternative methods reduce the problem to an RL objective with limit-average rewards instead of the standard discounted setting (Kazemi et al., 2022; Le et al., 2024). A variety of works also consider $LTL_f$ (De Giacomo & Vardi, 2013) or similar specification languages over *finite* traces, such as reward machines (Toro Icarte et al., 2018b; De Giacomo et al., 2018; 2019; Jothimurugan et al., 2019; 2021; Toro Icarte et al., 2022).

These approaches all consider only a *single*, fixed LTL specification, i.e. they learn a policy that maximises the probability of satisfying a given formula $\varphi$. In contrast, our approach is realised in a multi-task RL setting: we focus on learning a task-conditional policy that can zero-shot execute arbitrary LTL specifications at test time.

**Generalising to multiple tasks.** Toro Icarte et al. (2018a) were among the first to consider the problem of training a policy to complete multiple different tasks expressed in LTL. They propose a hierarchical algorithm based on $Q$-learning, which composes policies trained on subtasks. However, their approach is not able to generalise to novel formulae at test time, since these might consist of subtasks that the agent has not seen during training. Kuo et al. (2020) instead propose to leverage goal-conditioned RL with a compositional RNN architecture that consists of one RNN for every element in the syntax tree of a given LTL formula. While this method is shown to be able to generalise to tasks outside of the training distribution, it requires learning a non-stationary policy, which is known to be challenging (Vaezipoor et al., 2021).

León et al. (2021) introduce a different method for tasks expressed in a sub-fragment of $LTL_f$. They first employ a reasoning module to extract propositions that make progress towards solving the given task. These propositions are then achieved by a trained goal-conditioned policy. Similarly, Liu et al. (2024) propose a transfer algorithm that first trains a number of options on a set of training instructions, and then composes them at test time to achieve novel tasks. However, as noted by Vaezipoor et al. (2021) these approaches are inherently *myopic*: they do not take future propositions into account when executing the next subtask, and hence can produce suboptimal solutions. Instead, Vaezipoor et al. (2021) propose to directly encode the syntax tree of a given LTL formula using a GNN and predict actions based on the learned representations. To deal with the non-Markovian nature of LTL, they employ *LTL progression* (Bacchus & Kabanza, 2000). A more direct modification to the work of León et al. (2021) is proposed by Xu & Fekri (2024), who train *future-dependent* options for every proposition. These option policies are conditioned not only on the proposition to be achieved next, but also on the remaining propositions that need to be satisfied in the future. Qiu et al. (2023) introduce the first approach that can handle $\omega$-regular specifications. Their technique is based on training goal-conditioned policies $\pi(\cdot|s, p)$ to achieve arbitrary propositions $p \in AP$ in the environment. At test time, they employ a planning procedure to select a sequence of propositions to satisfy and finally execute the according low-level policies.

Our approach differs in a variety of ways from these previous methods: we leverage the structure of Büchi automata to find possible ways of satisfying a given specification. By operating on Büchi automata, our method can naturally handle $\omega$-regular specifications, which only GCRL-LTL is also capable of. However, in comparison to GCRL-LTL our method is non-myopic, since it incorporates the temporally extended structure of tasks via sequences of reach-avoid assignments. Additionally, compared to other methods that employ a high-level planning procedure (Qiu et al., 2023; Xu & Fekri, 2024) our approach considers safety requirements when selecting the optimal reach-avoid sequence, yielding plans that are more likely to be able to be executed without safety violations by the policy.

**Automata-conditioned RL.** A recent line of work related to LTL objectives is *automata-conditioned RL* (Yalcinkaya et al., 2023; 2024). Instead of using LTL to specify tasks, this line of research directly employs (compositions of) deterministic finite automata (DFAs). Yalcinkaya et al. (2024) explore this setting in a multi-task RL context, and propose to train a goal-conditioned policy conditioned on learned embeddings of DFAs. They also introduce the class of *reach-avoid derived* (RAD) DFAs, which they use for pre-training the automata embeddings.

Compared to our approach, automata-conditioned RL is strictly less expressive, since every DFA can trivially be represented as an LDBA (including objectives not expressible by LTL (Cohen-Chesnot, 1991)), but the converse is not true. In particular, DFAs cannot capture infinite-horizon tasks. Additionally, in contrast to RAD-embeddings (Yalcinkaya et al., 2024), we do not compute an embedding for the entire automaton, but instead extract satisfying reach-avoid sequences and select the optimal one according to the learned value function. This separates high-level reasoning (how to satisfy the specification) from low-level reasoning (how to act in the MDP) and allows the goal-conditioned policy to focus on achieving one particular sequence of propositions. We experimentally compare our approach to RAD-embeddings in Appendix G.8.

## E  COMPUTING ACCEPTING CYCLES

The algorithm to compute paths to accepting cycles is listed in Algorithm 1. The algorithm is based on depth-first search and keeps track of the currently visited path in order to extract all possible paths to accepting cycles.

---

**Algorithm 1** Computing paths to accepting cycles

---

**Require:**
  An LDBA $B = (\mathcal{Q}, q_0, \Sigma, \delta, \mathcal{F}, \mathcal{E})$ and current state $q$.
1: **procedure** DFS($q, p, i$)        ▷ $i$ is in the index of the last seen accepting state, or $-1$ otherwise
2:     $P \leftarrow \emptyset$
3:     **if** $q \in \mathcal{F}$ **then**
4:         $i \leftarrow |p|$
5:     **end if**
6:     **for all** $a \in 2^{AP} \cup \{\varepsilon\}$ **do**
7:         $p' \leftarrow [p, q]$
8:         $q' \leftarrow \delta(q, a)$
9:         **if** $q' \in p$ **then**
10:             **if** index of $q'$ in $p \leq i$ **then**
11:                 $P = P \cup \{p'\}$
12:             **end if**
13:         **else**
14:             $P = P \cup \text{DFS}(q', p', i)$
15:         **end if**
16:     **end for**
17:     **return** $P$
18: **end procedure**
19: $i \leftarrow 0$ if $q \in \mathcal{F}$ else $i \leftarrow -1$
20: **return**  DFS($q, [], i$)

---

## F  EXPERIMENTAL DETAILS

### F.1  ENVIRONMENTS

**LetterWorld.**  The *LetterWorld* environment has been introduced by Vaezipoor et al. (2021). It is a $7 \times 7$ grid world that contains 12 randomly placed letters corresponding to atomic propositions. Each letter appears twice, i.e. 24 out of the 49 squares are occupied, and there are thus multiple ways of solving any given task. The agent observes the full grid from an egocentric view. At each step, the agent can move up, right, down, or left. If it moves out of bounds, the agent is immediately placed on the opposite end of the grid. See Figure 8a for an illustration of the *LetterWorld* environment.

**ZoneEnv.**  We adapt the *ZoneEnv* environment introduced by Vaezipoor et al. (2021). The environment is a walled plane with 8 circular regions ("zones") that have four different colours and form the atomic propositions. Our implementation is based on the Safety Gymnasium suite (Ji et al., 2023) and uses the *Point* robot, which has a continuous action space for acceleration and steering. The environment features a high-dimensional state space based on lidar information about the zones, and

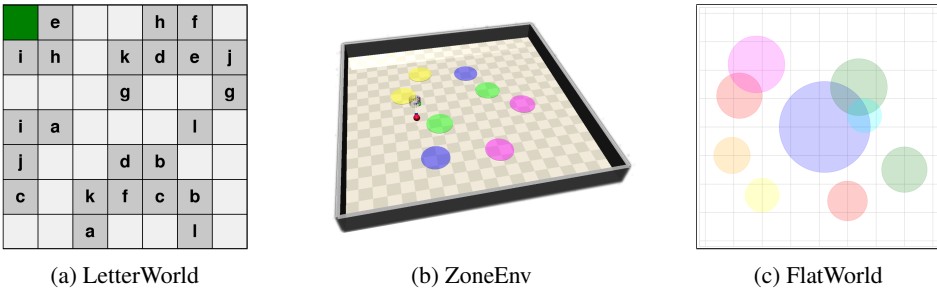

| (a) LetterWorld | (b) ZoneEnv | (c) FlatWorld |

Figure 8: Visualisations of environments.

data from other sensors. Both the zone and robot positions are randomly sampled at the beginning of each episode. If the agent at any point touches a wall, it receives a penalty and the episode is immediately terminated. A visualisation of the *ZoneEnv* environment is provided in Figure 8b.

**FlatWorld.** The *FlatWorld* environment (Voloshin et al., 2023; Shah et al., 2024) consists of a two-dimensional continuous world ($\mathcal{S} = [-2, 2]^2$) with a discrete action space. Atomic propositions are given by various coloured regions. Importantly, these regions overlap in various places, which means that multiple propositions can hold true at the same time. The initial agent position is sampled randomly from the space in which no propositions are true. At each time step, the agent can move in one of the 8 compass directions. If it leaves the boundary of the world, the agent receives a penalty and the episode is terminated prematurely. Figure 8c shows a visualisation of the *FlatWorld* environment.

### F.2 Testing specifications

Tables 2 and 3 list the finite and infinite-horizon specifications used in our evaluation, respectively.

### F.3 Hyperparameters

**Neural network architectures.** Our choice of neural network architectures is similar to previous work (Vaezipoor et al., 2021). For DeepLTL and LTL2Action, we employ a fully connected actor network with [64, 64, 64] units and ReLU as the activation function. The critic has network structure [64, 64] and uses Tanh activations in *LetterWorld* and *ZoneEnv*, and ReLU activations in *FlatWorld*. The actor is composed with a softmax layer in discrete action spaces, and outputs the mean and standard deviation of a Gaussian distribution in continuous action spaces. GCRL-LTL uses somewhat larger actor and critic networks with structure [512, 1024, 256] and ReLU activations in the *ZoneEnv* environment.

The observation module is environment-specific. For the *ZoneEnv* and *FlatWorld* environments, it consists of a simple fully connected network with [128, 64] units and Tanh activations, or [16, 16] units and ReLU activations, respectively. GCRL-LTL instead uses a simple projection of the input to dimensionality 100 in *ZoneEnv*. For *LetterWorld*, the observation module is a CNN with 16, 32, and 64 channels in three hidden layers, a kernel size of $2 \times 2$, stride of 1, no padding, and ReLU activations.

Finally, the sequence module consists of learned embeddings $\phi$ of dimensionality 32 in *LetterWorld*, and 16 in *ZoneEnv* and *FlatWorld*. The non-linear transformation $\rho$ is a fully connected network with [32, 32] units in *LetterWorld*, and [32, 16] units in *ZoneEnv* and *FlatWorld*. We use ReLU activations throughout for the sequence module. For the baselines, we use the hyperparameters reported in the respective papers (Vaezipoor et al., 2021; Qiu et al., 2023).

**PPO hyperparameters.** The hyperparameters for PPO (Schulman et al., 2017) are listed in Table 4. We use the Adam optimiser (Kingma & Ba, 2015) for all methods and environments.

**Additional hyperparameters.** LTL2Action requires an LTL task sampler to sample a random training specification at the beginning of each episode. We follow Vaezipoor et al. (2021) and

Table 2: *Complex* finite-horizon specifications used in our evaluation.

| | | |
|---|---|---|
| **LetterWorld** | $\varphi_1$ | $F(a \wedge (\neg b \ U \ c)) \wedge F \ d$ |
| | $\varphi_2$ | $F \ d \wedge (\neg f \ U \ (d \wedge F \ b))$ |
| | $\varphi_3$ | $F((a \vee c \vee j) \wedge F \ b) \wedge F(c \wedge F \ d) \wedge F \ k$ |
| | $\varphi_4$ | $\neg a \ U \ (b \wedge (\neg c \ U \ (d \wedge (\neg e \ U \ f))))$ |
| | $\varphi_5$ | $((a \vee b \vee c \vee d) \Rightarrow F(e \wedge (F(f \wedge F \ g)))) \ U \ (h \wedge F \ i)$ |
| **ZoneEnv** | $\varphi_6$ | $F(green \wedge (\neg blue \ U \ yellow)) \wedge F \ magenta$ |
| | $\varphi_7$ | $F \ blue \wedge (\neg blue \ U \ (green \wedge F \ yellow))$ |
| | $\varphi_8$ | $F(blue \vee green) \wedge F \ yellow \wedge F \ magenta$ |
| | $\varphi_9$ | $\neg(magenta \vee yellow) \ U \ (blue \wedge F \ green)$ |
| | $\varphi_{10}$ | $\neg green \ U \ ((blue \vee magenta) \wedge (\neg green \ U \ yellow))$ |
| | $\varphi_{11}$ | $((green \vee blue) \Rightarrow (\neg yellow \ U \ magenta)) \ U \ yellow$ |
| **FlatWorld** | $\varphi_{12}$ | $F((red \wedge magenta) \wedge F((blue \wedge green) \wedge F \ yellow))$ |
| | $\varphi_{13}$ | $F(orange \wedge (\neg red \ U \ magenta))$ |
| | $\varphi_{14}$ | $(\neg red \ U \ (green \wedge blue \wedge aqua)) \wedge F(orange \wedge (F(red \wedge magenta)))$ |
| | $\varphi_{15}$ | $((\neg yellow \wedge \neg orange) \ U \ (green \wedge blue)) \wedge (\neg green \ U \ magenta)$ |
| | $\varphi_{16}$ | $(blue \Rightarrow F \ magenta) \ U \ (yellow \vee ((green \wedge blue) \wedge F \ orange))$ |

Table 3: Infinite-horizon specifications used in our evaluation.

| | | |
|---|---|---|
| **LetterWorld** | $\psi_1$ | $G \ F(e \wedge (\neg a \ U \ f))$ |
| | $\psi_2$ | $G \ F \ a \wedge G \ F \ b \wedge G \ F \ c \wedge G \ F \ d \wedge G(\neg e \wedge \neg f)$ |
| **ZoneEnv** | $\psi_3$ | $G \ F \ blue \wedge G \ F \ green$ |
| | $\psi_4$ | $G \ F \ blue \wedge G \ F \ green \wedge G \ F \ yellow \wedge G \ \neg magenta$ |
| **FlatWorld** | $\psi_5$ | $G \ F(blue \wedge green) \wedge G \ F(red \wedge magenta)$ |
| | $\psi_6$ | $G \ F(aqua \wedge blue) \wedge G \ F \ red \wedge G \ F \ yellow \wedge G \ \neg green$ |

Table 4: Hyperparameters for PPO. Dashes (—) indicate that the hyperparameter value is the same across all three methods.

| | | LTL2Action | GCRL-LTL | DeepLTL |
|---|---|---|---|---|
| **LetterWorld** | Number of processes | — | 16 | — |
| | Steps per process per update | — | 128 | — |
| | Epochs | — | 8 | — |
| | Batch size | — | 256 | — |
| | Discount factor | — | 0.94 | — |
| | GAE-$\lambda$ | — | 0.95 | — |
| | Entropy coefficient | — | 0.01 | — |
| | Value loss coefficient | — | 0.5 | — |
| | Max gradient norm | — | 0.5 | — |
| | Clipping ($\epsilon$) | — | 0.2 | — |
| | Adam learning rate | — | 0.0003 | — |
| | Adam epsilon | — | 1e-08 | — |
| **ZonesEnv** | Number of processes | — | 16 | — |
| | Steps per process per update | 4096 | 3125 | 4096 |
| | Epochs | — | 10 | — |
| | Batch size | 2048 | 1000 | 2048 |
| | Discount factor | — | 0.998 | — |
| | GAE-$\lambda$ | — | 0.95 | — |
| | Entropy coefficient | — | 0.003 | — |
| | Value loss coefficient | — | 0.5 | — |
| | Max gradient norm | — | 0.5 | — |
| | Clipping ($\epsilon$) | — | 0.2 | — |
| | Adam learning rate | — | 0.0003 | — |
| | Adam epsilon | — | 1e-08 | — |
| **FlatWorld** | Number of processes | — | 16 | — |
| | Steps per process per update | — | 4096 | — |
| | Epochs | — | 10 | — |
| | Batch size | — | 2048 | — |
| | Discount factor | — | 0.98 | — |
| | GAE-$\lambda$ | — | 0.95 | — |
| | Entropy coefficient | — | 0.003 | — |
| | Value loss coefficient | — | 0.5 | — |
| | Max gradient norm | — | 0.5 | — |
| | Clipping ($\epsilon$) | — | 0.2 | — |
| | Adam learning rate | — | 0.0003 | — |
| | Adam epsilon | — | 1e-08 | — |

Table 5: Evaluation results of trained policies on *persistence* tasks. We report the average number of time steps for which the policy successfully remains in the target region after executing the $\varepsilon$-action. Results are averaged over 5 seeds and 500 episodes per seed. "$\pm$" indicates the standard deviation over seeds.

| | GCRL-LTL | DeepLTL |
|---|---|---|
| F G blue | $265.53_{\pm 94.54}$ | $\mathbf{562.81}_{\pm 136.28}$ |
| F G blue $\wedge$ F (yellow $\wedge$ F green) | $178.12_{\pm 62.88}$ | $\mathbf{336.81}_{\pm 069.43}$ |
| F G magenta $\wedge$ G $\neg$yellow | $406.52_{\pm 75.22}$ | $\mathbf{587.98}_{\pm 123.63}$ |
| G ((green $\vee$ yellow) $\Rightarrow$ F blue) $\wedge$ F G (green $\vee$ magenta) | $380.49_{\pm 84.74}$ | $\mathbf{570.37}_{\pm 138.98}$ |

sample specifications from the space of *reach/avoid* tasks. We also experimented with sampling more complex specifications, but found this detrimental to performance. For GCRL-LTL, we set the value threshold $\sigma$ to 0.9 in the *ZoneEnv* environment, and 0.92 in *LetterWorld* and *FlatWorld*. The threshold $\lambda$ for strict negative assignments in DeepLTL is set to 0.4 across experiments.

### F.4    TRAINING CURRICULA

We design training curricula in order to gradually expose the policy to more challenging tasks. The general structure of the curricula is the same across environments: we start with simple and short reach-avoid sequences, and move to more complicated sequences once the policy achieves satisfactory performance.

**LetterWorld.**    In the *LetterWorld* environment, the first curriculum stage consists of reach-avoid tasks of depth 1 with single propositions, e.g. $(\{\{a\}\}, \{\{b\}\}\})$. Once the policy achieves an average satisfaction rate of $95\%$ on these sequences, we move to the next curriculum stage, in which the depth is still 1, and $|A_1^+| \leq 2, |A_1^-| \leq 2$. The next stage consists of the same tasks with a length of 2. The final curriculum stage consists of length-3 sequences with $|A_i^+| \leq 2, |A_i^-| \leq 3$.

**ZoneEnv.**    For *ZoneEnv*, the first stages consist of first only reach-sequences of length up to 2 (i.e. $A_i^- = \emptyset$) and then reach-avoid sequences of length up to 2. We then increase the cardinality of the positive and negative assignments, while introducing sequences aligned with reach-stay tasks, e.g.

$$\Big( \Big( \{\{\text{green}\}\}, 2^{AP} \setminus \{\text{green}\} \Big), \dots \Big).$$

**FlatWorld.**    In *FlatWorld*, we first sample a mixture of reach- and reach-avoid sequences of depth up to 2. Once the policy achieves a success rate of $80\%$, we increase the cardinality of the positive and negative assignments up to 2.

## G    FURTHER EXPERIMENTAL RESULTS

### G.1    RESULTS ON PERSISTENCE TASKS

*Persistence* (a.k.a. *reach-stay*) tasks of the form F G a specify that a proposition needs to be true forever from some point on. We evaluate our approach on these tasks in the *ZoneEnv* environment, which features a continuous action space and is thus particularly challenging for reach-stay specifications. Table 5 compares the performance of our approach to the relevant baseline GCRL-LTL, where we report the average number of steps for which the agent successfully remains in the target region after executing the $\varepsilon$-action as the performance metric.[3] The results confirm that our method can successfully handle complex persistence tasks, and performs better than the baseline.

---

[3]When the policy executes the $\varepsilon$-action, this indicates that the target proposition should from then on be true forever (see e.g. $q_2$ in Figure 2). Since GCRL-LTL does not explicitly model the $\varepsilon$-actions required for F G specifications, we employ an approximation and execute the $\varepsilon$-action upon entering the target region for the first time, i.e. when the agent first enters the blue region for the task F G blue.

Table 6: Evaluation results of trained policies on long reachability (R-12) and reach-avoid (RA-6) tasks. We report the *success rate* (SR) and average number of steps to satisfy the task ($\mu$). Results are averaged over 5 seeds and 500 episodes per seed. "$\pm$" indicates the standard deviation over seeds.

|  | LTL2Action | | GCRL-LTL | | DeepLTL | |
|---|---|---|---|---|---|---|
|  | SR ($\uparrow$) | $\mu$ ($\downarrow$) | SR ($\uparrow$) | $\mu$ ($\downarrow$) | SR ($\uparrow$) | $\mu$ ($\downarrow$) |
| R-12 | $0.89_{\pm 0.14}$ | $47.80_{\pm 7.74}$ | $0.93_{\pm 0.05}$ | $60.15_{\pm 2.59}$ | $\mathbf{0.98}_{\pm 0.01}$ | $\mathbf{43.33}_{\pm 0.45}$ |
| RA-6 | $0.13_{\pm 0.03}$ | $56.37_{\pm 0.89}$ | $0.62_{\pm 0.09}$ | $30.64_{\pm 1.59}$ | $\mathbf{0.95}_{\pm 0.01}$ | $\mathbf{24.94}_{\pm 0.36}$ |

## G.2 ADVERSARIAL TASKS WITH SAFETY CONSTRAINTS

We next demonstrate the advantages of our approach in terms of handling safety constraints on an adversarial task. This task is specifically designed to require considering safety requirements during high-level planning. We consider the configuration of the *ZoneEnv* environment depicted in Figure 9 and the specification $\varphi = \neg\mathsf{blue}\ \mathsf{U}\ (\mathsf{green} \vee \mathsf{yellow}) \wedge \mathsf{F}\ \mathsf{magenta}$.

Figure 9 shows a number of sample trajectories generated by DeepLTL (top row) and GCRL-LTL (bottom row). Evidently, GCRL-LTL fails at satisfying the task, since it decides on reaching the green region during high-level planning without considering the safety constraint $\neg\mathsf{blue}$. In contrast, DeepLTL considers this safety requirement when selecting the optimal reach-avoid sequence, and hence chooses to reach the yellow region instead.

Quantitatively, GCRL-LTL only succeeds in satisfying the task in $9.2\%$ of cases, whereas DeepLTL achieves a success rate of $79.6\%$ (averaged over 5 seeds and 100 episodes per seed). While LTL2Action does not have a high-level planning component, it considers the entire task including safety constraints via its GNN encoding, and averages a success rate of $50.2\%$.

## G.3 GENERALISATION TO LONGER SEQUENCES

In this section, we investigate the ability of our approach to generalise to longer sequences. Our evaluation so far has already demonstrated that our method can successfully learn general behaviour for satisfying LTL specifications by only training on simple reach-avoid sequences. We now extend our analysis by specifically investigating tasks that require a large number of steps to solve.

In particular, we consider the following two task spaces in the *LetterWorld* environment: sequential reachability objectives of depth 12 (i.e. $\mathsf{F}\ (p_1 \wedge (\mathsf{F}\ p_2 \wedge \ldots \wedge \mathsf{F}\ p_{12}))$) and reach-avoid specifications of depth 6 (i.e. $\neg p_1\ \mathsf{U}\ (p_2 \wedge \ldots \wedge (\neg p_{11}\ \mathsf{U}\ p_{12})))$. Note that the longest reach-avoid sequences sampled during training are of length 3. In Table 6, we report the results of our method and the baselines on randomly sampled tasks from the task spaces described above. We observe that our approach generalises well to longer sequences and outperforms the baselines in terms of satisfaction probability and efficiency.

## G.4 ABLATION STUDY: CURRICULUM LEARNING

We conduct an ablation study in the *LetterWorld* environment to investigate the impact of curriculum learning. We train our method without any training curriculum by directly sampling random reach-avoid sequences of length 3, with potentially multiple propositions to reach and avoid at each stage. The evaluation curves on *reach/avoid* specifications over training (Figure 10) demonstrate that curriculum training improves sample efficiency and reduces variance across seeds.

## G.5 ABLATION STUDY: SEQUENCE MODULE

We conduct a further ablation study to investigate the impact of replacing the RNN in the sequence module with a Transformer (Vaswani et al., 2017) model. Since the reach-avoid sequences arising from common LTL tasks are generally relatively short (2-15 tokens), we expect the simplicity and effectiveness of RNNs to outweigh the benefits of the more complicated Transformer architecture in our setting. For our ablation study, we use a Transformer encoder to learn an embedding of the reach-avoid sequence as the final representation of a special [CLS] token, and keep the rest of the

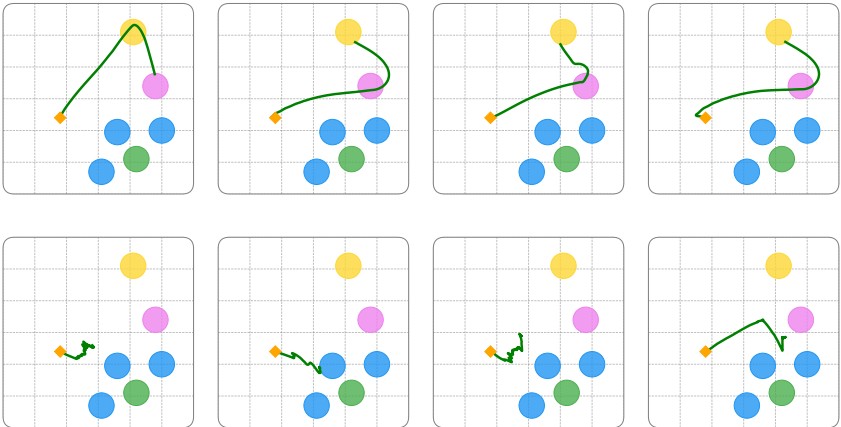

Figure 9: Trajectories of DeepLTL (top row) and GCRL-LTL (bottom row) on the adversarial configuration of the *ZoneEnv* environment. The specification to be completed is $\varphi = \neg$blue U (green $\vee$ yellow) $\wedge$ F magenta. GCRL-LTL ignores safety constraints during high-level planning, and hence performs poorly. In contrast, DeepLTL yields trajectories that satisfy the specification.

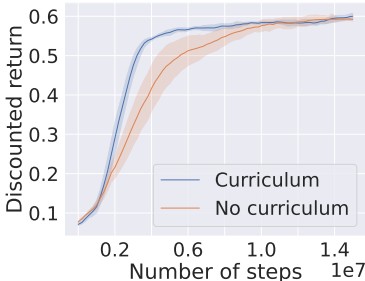

Figure 10: Evaluation curves of training with and without a curriculum on the *LetterWorld* environment. Each datapoint is collected by averaging the discounted return of the policy across 50 episodes with randomly sampled *reach/avoid* specifications, and shaded areas indicate 90% confidence intervals over 5 different random seeds.

Table 7: Comparison of Transformer- and GRU-based sequence modules. We report the *success rate* (SR) and average number of steps to satisfy the task ($\mu$). Results are averaged over 5 seeds and 500 episodes per seed. "$\pm$" indicates the standard deviation over seeds.

| | DeepLTL-Transformer | | DeepLTL-GRU | |
|---|---|---|---|---|
| | SR ($\uparrow$) | $\mu$ ($\downarrow$) | SR ($\uparrow$) | $\mu$ ($\downarrow$) |
| $\varphi_6$ | $0.78_{\pm 0.09}$ | $462.03_{\pm 52.60}$ | $\mathbf{0.92}_{\pm 0.06}$ | $\mathbf{303.38}_{\pm 19.43}$ |
| $\varphi_7$ | $0.46_{\pm 0.04}$ | $438.24_{\pm 42.35}$ | $\mathbf{0.91}_{\pm 0.03}$ | $\mathbf{299.95}_{\pm 09.47}$ |
| $\varphi_8$ | $0.93_{\pm 0.03}$ | $357.34_{\pm 51.86}$ | $\mathbf{0.96}_{\pm 0.04}$ | $\mathbf{259.75}_{\pm 08.07}$ |
| $\varphi_9$ | $0.74_{\pm 0.02}$ | $275.11_{\pm 31.56}$ | $\mathbf{0.90}_{\pm 0.03}$ | $\mathbf{203.36}_{\pm 14.97}$ |
| $\varphi_{10}$ | $0.80_{\pm 0.15}$ | $260.58_{\pm 29.48}$ | $\mathbf{0.91}_{\pm 0.02}$ | $\mathbf{187.13}_{\pm 10.61}$ |
| $\varphi_{11}$ | $0.90_{\pm 0.07}$ | $137.46_{\pm 21.26}$ | $\mathbf{0.98}_{\pm 0.01}$ | $\mathbf{106.21}_{\pm 07.88}$ |

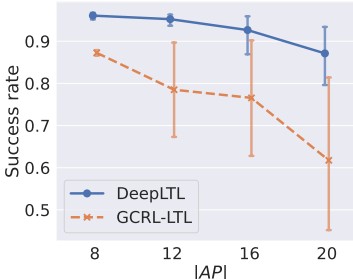 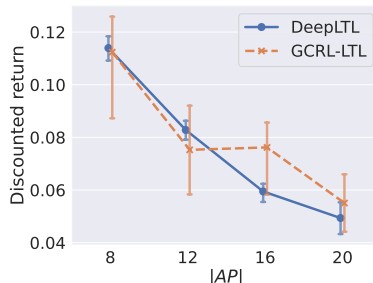

Figure 11: Impact of varying the number of atomic propositions on success rate (left) and discounted return (right). Each datapoint is an average over 500 episodes with randomly sampled *reach/avoid* specifications, and error bars indicate 90% confidence intervals over 5 different random seeds.

model architecture the same. We use an encoder model with an embedding size of 32, 4 attention heads, and a 512-dimensional MLP with dropout of 0.1. Despite this being a very small Transformer model, it still has more than 5 times the number of parameters of the GRU.

We report evaluation results for *complex* specifications in the *ZoneEnv* environment in Table 7, which strongly support our hypothesis that a simpler RNN-based sequence module is more effective.

### G.6 IMPACT OF THE NUMBER OF ATOMIC PROPOSITIONS

We conduct an experiment to evaluate the impact of the number of atomic propositions. For this, we consider modified versions of the *FlatWorld* environment with $n_r \in \{8, 12, 16, 20\}$ regions of different colours (that is, in each version we have $n_r$ regions and $|AP| = n_r$). The state space of the *FlatWorld* environment does not depend on the atomic propositions, therefore we can use the same model architecture throughout the experiment.

The results are shown in Figure 11. We observe that policy performance generally decreases as the number of atomic propositions increases, both for DeepLTL and GCRL-LTL. This result is expected, since we increase the number of possible tasks while keeping the number of policy parameters fixed. We also observe that the variance of success becomes larger as the number of atomic propositions increases. This can be explained by the fact that, in the case of a large number of atomic propositions, the goal-conditioned RL process may (for some random seeds) yield a policy that cannot consistently satisfy every single proposition. If there exists a proposition $a \in AP$ that the policy has failed to learn how to satisfy, the policy will consequently not be successful in completing any task that requires $a$ to be achieved at any stage, resulting in a low success rate. Finally, we observe that the performance reduction for DeepLTL generally is less than or similar to the performance reduction for GCRL-LTL.

### G.7 IMPACT OF SPECIFICATION COMPLEXITY

In this section, we provide additional insights regarding the performance of our approach with respect to the complexity of the given specification. An advantage of our method is the representation of tasks as reach-avoid sequences. This representation explicitly lists the possible ways of satisfying a specification, making our approach in principle applicable to arbitrarily complex formulae. As an example, we consider the following specification in the *LetterWorld* environment:

$$(a \Rightarrow F(b \wedge (F(c \wedge Fd)))) U \big((F((k \vee b) \wedge ((\neg b \wedge \neg d \wedge \neg j) U l))) \wedge (\neg h U k)\big) \vee$$
$$F\big(i \wedge (((b \vee c \vee k) \Rightarrow F(j \wedge Fi)) U ((l \vee f) \wedge (\neg(b \vee c \vee d) U g)))\big),$$

which encompasses 11 unique atomic propositions and a complex composition of different LTL operators. The corresponding LDBA consists of a large number of 656 states. However, by explicitly reasoning about the different ways of satisfying the specification, DeepLTL achieves an average success rate of 98% (across 5 different random seeds).

At the same time, we note that even seemingly simple formulae can be difficult to accomplish. In *LetterWorld*, this is especially true if they involve a large number of propositions to avoid, such as the specification $\neg(a \vee b \vee c \vee d \vee e) U (f \wedge Fg)$, on which DeepLTL achieves a comparatively low

Table 8: Experimental comparison to RAD-embeddings on different finite-horizon tasks in the *ZoneEnv* environment. The numbers in parentheses represent the number of states in the LDBA/DFA. We report the *success rate* (SR) and average number of steps to satisfy the task ($\mu$). Results are averaged over 5 seeds and 500 episodes per seed. "$\pm$" indicates the standard deviation over seeds.

| | RAD-embeddings | | DeepLTL | |
|---|---|---|---|---|
| | SR ($\uparrow$) | $\mu$ ($\downarrow$) | SR ($\uparrow$) | $\mu$ ($\downarrow$) |
| $\varphi_6$ (6) | $\mathbf{0.94}_{\pm 0.06}$ | $469.72_{\pm 59.99}$ | $0.92_{\pm 0.06}$ | $\mathbf{303.38}_{\pm 19.43}$ |
| $\varphi_7$ (6) | $0.88_{\pm 0.06}$ | $474.34_{\pm 62.14}$ | $\mathbf{0.91}_{\pm 0.03}$ | $\mathbf{299.95}_{\pm 09.47}$ |
| $\varphi_8$ (8) | $\mathbf{0.96}_{\pm 0.06}$ | $399.38_{\pm 68.63}$ | $\mathbf{0.96}_{\pm 0.04}$ | $\mathbf{259.75}_{\pm 08.07}$ |
| $\varphi_9$ (5) | $\mathbf{0.92}_{\pm 0.02}$ | $304.88_{\pm 38.42}$ | $0.90_{\pm 0.03}$ | $\mathbf{203.36}_{\pm 14.97}$ |
| $\varphi_{10}$ (4) | $0.87_{\pm 0.11}$ | $336.90_{\pm 88.32}$ | $\mathbf{0.91}_{\pm 0.02}$ | $\mathbf{187.13}_{\pm 10.61}$ |
| $\varphi_{11}$ (4) | $\mathbf{0.99}_{\pm 0.01}$ | $149.02_{\pm 19.72}$ | $0.98_{\pm 0.01}$ | $\mathbf{106.21}_{\pm 07.88}$ |
| *reach/avoid* (4/5) | $0.91_{\pm 0.04}$ | $401.12_{\pm 33.92}$ | $\mathbf{0.93}_{\pm 0.02}$ | $\mathbf{261.97}_{\pm 11.79}$ |
| *parity* (5) | $\mathbf{0.98}_{\pm 0.01}$ | $292.21_{\pm 27.27}$ | $0.96_{\pm 0.03}$ | $\mathbf{211.07}_{\pm 12.16}$ |
| *large1* (15) | $0.65_{\pm 0.32}$ | $550.51_{\pm 95.79}$ | $\mathbf{0.92}_{\pm 0.06}$ | $\mathbf{314.72}_{\pm 23.23}$ |
| *large2* (19) | $0.66_{\pm 0.31}$ | $490.22_{\pm 40.23}$ | $\mathbf{0.94}_{\pm 0.02}$ | $\mathbf{322.38}_{\pm 24.02}$ |

success rate of $82\%$ across 5 random seeds. In general, whether a given formula is challenging to satisfy depends primarily on the dynamics of the underlying MDP.

Finally, we note that our approach relies on being able to construct an LDBA for a given specification, which is doubly-exponential in the size of the formula in the worst case (Sickert et al., 2016). Inevitably, there thus exist LTL specifications to which our approach is not applicable, since it is infeasible to construct the corresponding LDBA.

## G.8 COMPARISON TO RAD-EMBEDDINGS

Table 8 lists experimental results of DeepLTL and RAD-embeddings (Yalcinkaya et al., 2024) on finite-horizon tasks in the *ZoneEnv* environment. We consider the tasks $\varphi_6 — \varphi_{11}$ from Table 2, *reach/avoid* tasks (see Section 5.1), as well as a *parity* task, and two tasks with large associated automata.[4][5] The *parity* task specifies that the agent has to reach blue while visiting green an even number of times, and yellow while visiting magenta an even number of times. Note that this task is not expressible in LTL, but our method can still be applied to the corresponding LDBA.

For RAD-embeddings, we use frozen GATv2 embeddings pre-trained on compositional RAD DFAs, and the policy is also trained on compositional RAD DFAs for 15M environment interaction steps. Throughout the experiments, we use the hyperparameters provided in (Yalcinkaya et al., 2024).

The results shows that RAD-embeddings and DeepLTL achieve similar success rates, but DeepLTL generally requires significantly fewer steps until task completion. They also demonstrate that our approach indeed generalises to tasks not expressible in LTL, such as *parity*. Finally, we note that our method achieves substantially higher success rates on tasks with an increased number of states in the corresponding automaton (*large1/2*). These results further support the advantages of our approach of combining a high-level reasoning step with learned embeddings of reach-avoid sequences.

## G.9 TRAJECTORY VISUALISATIONS

We qualitatively confirm that DeepLTL produces the desired behaviour by visualising trajectories in the *ZoneEnv* and *FlatWorld* environments for a variety of tasks (Figures 12 and 13).

---

[4]*large1* corresponds to the formula ((green $\Rightarrow$ ($\neg$blue U ((magenta $\wedge$ F blue) $\wedge$ ($\neg$green U blue))))) U (yellow $\wedge$ F (magenta $\wedge$ F blue))).

[5]The formula for *large2* is (green $\Rightarrow$ F (yellow $\wedge$ (F (magenta $\wedge$ F blue)))) U ((blue $\wedge$ F magenta) $\wedge$ ($\neg$yellow U green)).

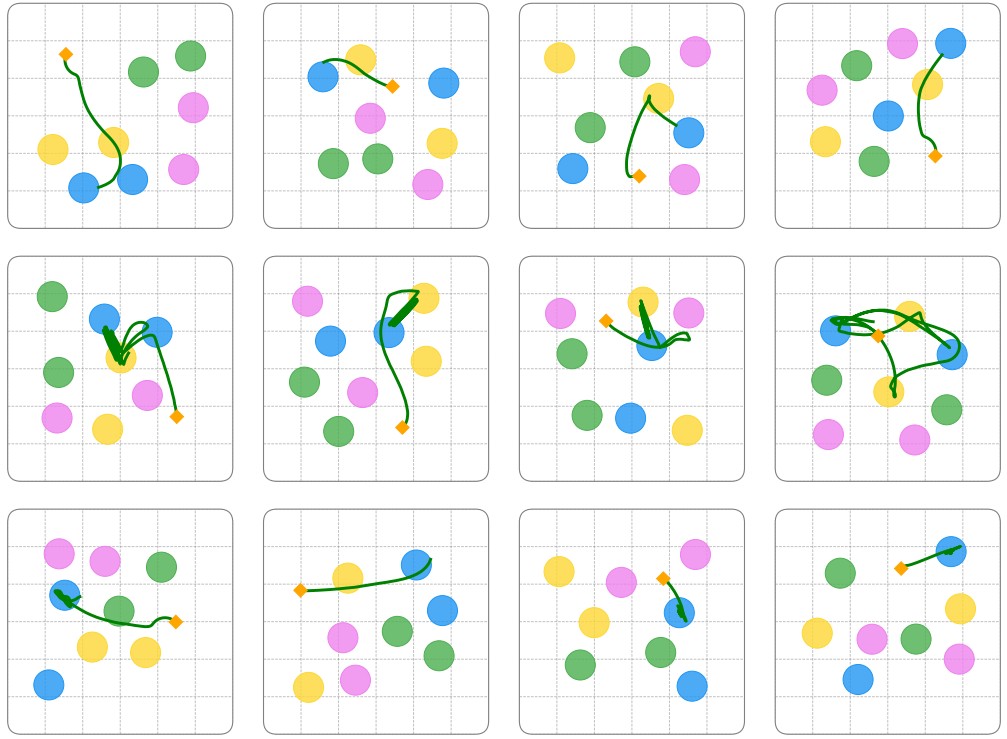

Figure 12: Example trajectories of DeepLTL in the *ZoneEnv* environment. (Top row) Trajectories for the task F (yellow ∧ (¬green U blue)). (Middle row) Trajectories for the task G F yellow ∧ G F blue ∧ G ¬green. (Bottom row) Trajectories for the task F G blue ∧ G ¬magenta.

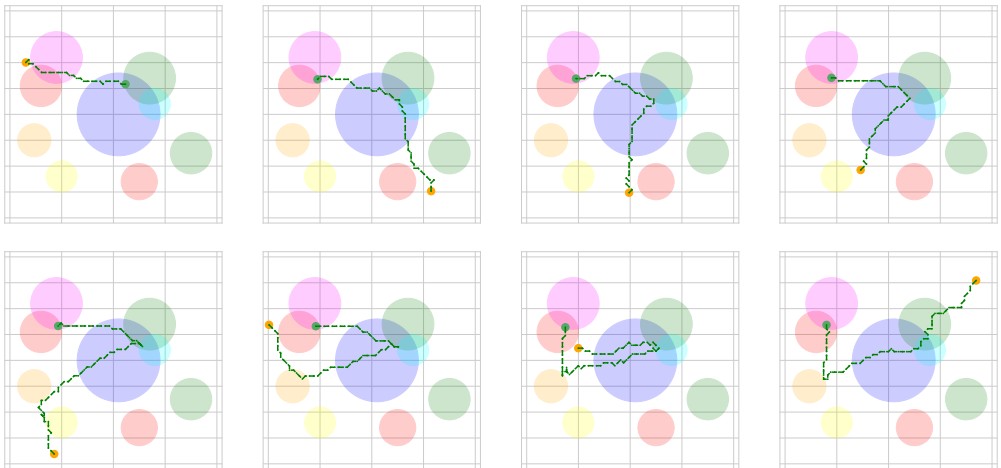

Figure 13: Example trajectories of DeepLTL in the *FlatWorld* environment. (Top row) Trajectories for the task F (red∧magenta)∧F (blue∧green). (Bottom row) Trajectories for the task (¬red U (green∧ blue ∧ aqua)) ∧ F (orange ∧ (F (red ∧ magenta))).

