# OpenReview forum: "DeepLTL: Learning to Efficiently Satisfy Complex LTL Specifications for Multi-Task RL"
_ICLR.cc/2025/Conference — ICLR 2025 Oral_

### Official Review · Reviewer_T576 · 2024-10-27

**Soundness:** 3
**Presentation:** 3
**Contribution:** 4
**Rating:** 8
**Confidence:** 4

**Summary:**

This paper presents a reinforcement learning based policy synthesis method for a robot to satisfy a Linear Temporal Logic (LTL) specification. The salient features that distinguish this paper from prior work are the following: (1) the proposed method does not aim to generate a policy for a fixed LTL formula but rather to deal with any arbitrary one, (2) it can deal with specifications that can be satisfied only through infinite length execution,  and (3) it ensures the satisfaction of the safety requirements, and (4) it optimizes the length of the trajectory. The proposed method is based on the observation that the satisfaction of a specification primarily depends on the loops including the final states in the Buchi automaton equivalent to the given specification. For a given LTL formula, the sequence of the sets of actions that lead to the satisfaction and violation of the specification is identified and the policy is trained based on those sequences. On the test time, the policy for the target LTL formula can utilize the policy learnt based on many different LTL specifications and thus the learnt policy can be used in a zero-shot manner. The authors evaluate their method on three benchmark environments and compare it with two baselines. Experimental results establish the proposed method to be superior to the state-of-the-art methods both in terms of the rate of success in satisfying the test specifications and the optimality of the length of the trajectories.

**Strengths:**

This paper improves the state-of-the-art for reinforcement learning with LTL specifications in several directions. Unlike the earlier methods, the proposed technique can deal with arbitrary LTL specifications at test time, supports infinite-horizon LTL specifications, ensures the satisfaction of the safety constraints, and attempts to optimize the trajectory length. Thus the technical contribution of the paper is significant.

The experimental evaluation is quite exhaustive, establishing the efficacy of the proposed method compared to the state-of-the-art.

**Weaknesses:**

The presentation in some parts of the paper could be improved. Specifically, a running example could help understand several complex ideas. For example, Section 4.2 could be easier to understand had an example been provided. Similarly, the paragraph on representing the reach-avoid sequence on page 6 could also be accompanied by an example. Furthermore, an example of how the negative assignments help could help convince readers about their necessity.

**Questions:**

In Example 1, why can’t we replace the transition on $\epsilon_{q_2}$ by a transition on the action $a$ to generate an equivalent Buchi automata?

In Line 252, in $\delta(q_i, a) \ne q_i$, wouldn’t the second $q_i$ be $q_{i+1}$?

Is it not the case that restricting the actions in the set $A_i^+$ will ensure that the actions are not from the sets $A_i^-$?  These two sets appear to be mutually exclusive. Then why do we need to keep track of both?

Some of the terms used in the paper have never been introduced. For example, what is $sup(\xi)$? How to interpret $\tau \sim \pi | \varphi$?

On Line 107, please use $\equiv$ instead of “=“ to denote formula equivalence.

**Details Of Ethics Concerns:**

No Concerns.

---

> ### Author Response · Authors · 2024-11-24
>
> Thank you for taking the time to review our paper! We are pleased to read your positive comments, especially that the "technical contribution of the paper is significant" and that we perform an "exhaustive" experimental evaluation. Please see below for answers and comments regarding the points raised.
>
> **Negative assignments $A_i^-$**
>
> We think there might be a small misunderstanding regarding the negative assignments. For a path $(q_1, q_2, \ldots)$ in the LDBA, the *positive* assignments $A_i^+ = \\{ a : \delta(q_i, a) = q_{i+1} \\}$ are the assignments that lead to the next state $q_{i+1}$. The *negative* assignments are all assignments that do not lead to $q_{i+1}$ and *do not form a self-loop*. This is why we have the condition $\delta(q_i,a)\neq q_i$ in the definition of the set $A_i^-$.
>
> We appreciate that this might not have been entirely clear in the writing. We have thus made a small change to Section 4.1, explicitly mentioning that $A_i^-$ excludes self-loops, to hopefully make this clearer.
>
> > Is it not the case that restricting the actions in the set $A_i^+$ will ensure that the actions are not from the sets $A_i^-$?  These two sets appear to be mutually exclusive. Then why do we need to keep track of both?
>
> This question relates to our discussion above. $A_i^-$ contains assignments that the policy needs to avoid, i.e. assignments that lead to a different state than the desired one and that do not form a self-loop. For example, consider the formula $\neg a \mathsf{U} b$. In this case, $A_i^-$ contains all assignments where $a$ is true (since this leads to an undesired state in the automaton), $A_i^+$ contains all assignments in which $b$ is true, and all other assignments can safely be ignored by the policy since they keep it in the same LDBA state (e.g. the assignment $\\{ a\mapsto \text{false}, b\mapsto\text{false}, c\mapsto\text{true} \\}$). Intuitively, the policy's goal is to materialise an assignment in $A_i^+$, but this may require many steps, during which it must avoid assignments in $A_i^-$ but is allowed to materialise other assignments.
>
> **Comments and questions**
>
> > Section 4.2 could be easier to understand had an example been provided. Similarly, the paragraph on representing the reach-avoid sequence on page 6 could also be accompanied by an example.
>
> Many thanks for this suggestion. Since we already provide high-level examples in Figure 3 and Figure 4, do you have any suggestions how we can improve them to make the relevant sections easier to understand?
>
> > In Example 1, why can’t we replace the transition on $\varepsilon_{q_2}$ by a transition on the action $a$ to generate an equivalent Buchi automata?
>
> You are right that we could in principle replace the $\varepsilon$-transition with an $a$-transition to obtain a Büchi automaton accepting the same language. However, the resulting automaton would not be an LDBA. In particular, note that the transition $\neg b$ (the self-loop on state $q_0$) already contains the assignment in which $a$ is true. As such, the transition function would be non-deterministic for the input $a$ in state $q_0$.
>
> The advantage of LDBAs is that they contain all non-determinism in the $\varepsilon$-transitions, which can be folded into the action space of the policy, and thus learned (see Definition 1 and the discussion thereafter). If we used a non-deterministic Büchi automaton, we would not know how to progress the product MDP in the case of a non-deterministic transition.
>
> > Some of the terms used in the paper have never been introduced. For example, what is $supp(\xi)$? How to interpret $\tau\sim\pi|\varphi$?
>
> $supp(\xi)$ denotes the *support* of probability distribution $\xi$, i.e. all formulae with nonzero probability. We have clarified this in the revised version of the paper. We introduce the notation $\tau\sim\pi$ in line 96 and the notation $\pi|\varphi$ for a specification-conditioned policy in line 147.
>
> > On Line 107, please use $\equiv$ instead of “=“ to denote formula equivalence.
>
> Many thanks, we have revised the paper accordingly.
>
> Thank you again for your comments and feedback! We hope our response and edits have clarified some of the points. Please let us know if you have any other questions or remarks!

---

> > ### Comment · Reviewer_T576 · 2024-11-27
> >
> > Thank you for your thorough response. I was already positive about the paper, and the additional clarifications in the rebuttal have further reinforced my confidence in its acceptability. I would like to congratulate the authors on their outstanding work.

---

### Official Review · Reviewer_7njW · 2024-11-04

**Soundness:** 3
**Presentation:** 3
**Contribution:** 3
**Rating:** 8
**Confidence:** 4

**Summary:**

The paper introduces a novel approach, called DeepLTL, to address the challenge of learning policies that ensure the satisfaction of arbitrary LTL specifications over an MDP. This approach reduces the myopic tendencies found in previous works by representing each specification as a set of reach-avoid sequences of truth assignments. It then leverages a general sequence-conditioned policy to execute arbitrary LTL instructions at test time. Extensive experiments demonstrate the practical effectiveness of this approach.

**Strengths:**

The proposed approach is tailored to address key challenges of quality, clarity, and significance. Unlike existing techniques, this method is designed to handle infinite-horizon specifications and mitigate the non-myopic tendencies of previous approaches that often lead to sub-optimality. Additionally, it naturally incorporates safety constraints, represented through negative assignments, to guide the policy on propositions to avoid, which is an essential concept for effective planning. In general, the paper is well-written and effectively presented.

**Weaknesses:**

The approach proposed by the authors is compelling and aims to address an important problem. However, one concern is that the authors appear unaware of works like [1], [2], and [3], which introduced model-free reinforcement learning (RL) methods to tackle the same challenge of maximizing the probability of satisfaction for LTL specifications, expressed as Büchi automata and deterministic parity automata. These methods have even been extended to nondeterministic, adversarial environments (expressed as stochastic games) where nonrandom actions are taken to disrupt task performance, beyond standard MDPs. In such approaches, the LTL specifications are translated into limit-deterministic Büchi automata (LDBAs) to form product MDPs. Rewards are derived from automata using a repeated reachability acceptance condition, allowing controller strategies that maximize cumulative discounted rewards to also maximize satisfaction probabilities; standard RL algorithms are then used to learn these strategies. In my opinion, these results appear to weaken the authors’ claim that ‘Our method is the first approach that is also non-myopic, as it is able to reason about the entire structure of a specification via temporally extended reach-avoid sequences.’ Please discuss how your approach compares to and differs from the methods in [1], [2], and [3], with particular attention to handling non-myopic reasoning and addressing infinite-horizon specifications.

A. K. Bozkurt, Y. Wang, M. M. Zavlanos, and M. Pajic, “Control synthesis from linear temporal logic specifications using model-free reinforcement learning,” in Proc. Int. Conf. Robot. Automat., 2020, pp. 10349–10355

E. M. Hahn, M. Perez, S. Schewe, F. Somenzi, A. Trivedi, and D. Wojtczak, “Omega-regular objectives in model-free reinforcement learning,” in Proc. Int. Conf. Tools Algorithms Construction Anal. Syst., 2019, pp. 395–412.

Learning Optimal Strategies for Temporal Tasks in Stochastic Games Alper Kamil Bozkurt , Yu Wang , Michael M. Zavlanos , and Miroslav Pajic

**Questions:**

The examples provided by the authors are all based on 2D grid-world environments. To evaluate the approach's performance in higher-dimensional settings, it would be valuable to experiment with environments like the 5-dimensional Carlo environment from [1], as well as other high-dimensional settings, such as the Fetch environment in [2], as utilized in [3]. Additionally, as a minor note, there is a typo on line 066 of the paper; it should read (c) instead of (b).

[1] Cameron Voloshin, Abhinav Verma, and Yisong Yue. Eventual Discounting Temporal Logic Counterfactual Experience Replay. In Proceedings of the 40th International Conference on Machine Learning, pp. 35137–35150. PMLR, July 2023.

[2] M. Plappert et al., “Multi-goal reinforcement learning: Challenging robotics environments and request for research,” 2018, arXiv:1802.09464.

[3] Learning Optimal Strategies for Temporal Tasks in Stochastic Games Alper Kamil Bozkurt , Yu Wang , Michael M. Zavlanos , and Miroslav Pajic

---

> ### Author Response · Authors · 2024-11-15
>
> We thank you for the detailed feedback and are pleased that you found our approach "compelling" and our paper "well-written and effectively presented". Nevertheless, we would like to point out some possible misunderstandings that might have negatively impacted the assessment.
>
> **Comparison to [1], [2], and [3]**
>
> We appreciate these references and agree that they are generally relevant in the context of our work. However, please note that these approaches tackle a **different/simpler problem** than the one we consider and are **not applicable** to our setting. In particular, our approach is realised in a **multi-task RL** setting and we train a single policy that can **zero-shot execute arbitrary unseen LTL specifications at test time**. In contrast, the methods in [1], [2], and [3] only learn a policy for a **single, fixed task**. This is a crucial difference: our approach trains a single policy once, which can then satisfy arbitrary tasks such as the ones in our evaluation (see Tables 2 and 3). The given references would have to train a separate new policy for every specification, and cannot generalise to new tasks at test time.
>
> We have submitted an updated version of the paper to hopefully make this distinction clearer. The updated version includes a changed title that explicitly mentions multi-task RL, and minor corresponding edits to the abstract and introduction. We have also updated the related work section (Section 6) to more explicitly point out the differences to methods that only handle a single task, and include an extended discussion of related work in the appendix (see Appendix C in the updated paper), which includes the provided references.
>
> > In my opinion, these results appear to weaken the authors’ claim that ‘Our method is the first approach that is also non-myopic [...]'
>
> We appreciate that previous methods that learn a policy for a single LTL specification are generally non-myopic and can handle infinite-horizon specifications. However, we are not aware of any non-myopic method that is able to satisfy infinite-horizon specifications in the multi-task RL setting. If you are aware of such a method, we would greatly appreciate a reference.
>
> **2D grid-world environments**
>
> Please note that our experiments include the **high-dimensional ZoneEnv** environment with **continuous state and action spaces**, which is a standard environment in previous research on multi-task RL with LTL tasks (Vaezipoor et al. 2021, Qiu et al. 2023). This is a Mujoco environment consisting of a robot navigating a planar world with continuous acceleration and steering actions while observing sensory information, including lidar observations about various coloured zones. The state-space of this environment is 80-dimensional (compared to the 25-dimensional Fetch environment and the 5-dimensional Carlo environment). For a description of the environment see Section 5.1 and Appendix E.
>
> We also consider the FlatWorld environment, which similarly features a continuous state space (albeit of lower dimensionality 2).
>
> We therefore believe our experiments already demonstrate the performance of our approach in high-dimensional and continuous environments.
>
> > Additionally, as a minor note, there is a typo on line 066 of the paper; it should read (c) instead of (b).
>
> Many thanks, we fixed the typo in the updated version of the paper.
>
> We appreciate your comments and hope that our response with the corresponding edits in the revised version of the paper has made our contribution clearer; in particular the difference to related work that handles only a single LTL specification, and our evaluation in high-dimensional environments. Please let us know if this mitigates your concerns!

---

> > ### Comment · Reviewer_7njW · 2024-11-17
> > **Updating my review score**
> >
> > I appreciate the authors' clarification on the contributions of their paper. The earlier presentation made it difficult to distinguish their contributions from previously established results. The referenced works focus on maximizing the satisfaction probability of a system over a single LTL objective, whereas this paper extends the problem to a probability distribution over a collection of LTL tasks. This extension broadens the applicability of the results to multi-task settings, leveraging a curriculum learning approach.
> >
> > As a result, this work contributes to multi-task reinforcement learning by enabling the training of a single policy capable of zero-shot execution of arbitrary, unseen LTL specifications at test time. Additionally, the improved clarity in the provided example effectively addresses my earlier concerns. The clear distinction in the scope of the contribution underscores the significance of the paper’s results. I, therefore, recommend its acceptance.

---

> > > ### Author Response · Authors · 2024-11-18
> > >
> > > We would like to thank the reviewer for reassessing our paper, and are pleased to hear that the updates have clarified the contributions of our work. Many thanks again for your feedback.

---

### Official Review · Reviewer_Xrc5 · 2024-11-04

**Soundness:** 3
**Presentation:** 3
**Contribution:** 2
**Rating:** 8
**Confidence:** 5

**Summary:**

The authors propose a multi-task RL approach using goals specified in Linear Temporal Logic. The approach builds on recent work by reasoning about *accepting cycles* in the form of reach-avoid sequences and learns a goal-conditioned policy that can generalize to unseen specifications by finding the highest-valued reach-avoid sequence in the new specification('s automata), where the reach-avoid sequence goals are cast as learned embeddings. The approach is trained in a multi-task setting with a simple curriculum, and experimental results demonstrate that the DeepLTL approach outperforms previous approaches to goal-conditioned LTL-modulo-RL.

**Strengths:**

* The paper is overall well-written and nicely constructed.
* The problem of multi-task RL is, in my opinion, one of the most salient applications of using structured logical specifications. I think the paper does a nice job of trying to extend this.
* The paper does a god job contextualizing some of the recent theory (e.g. regarding the eventual discounting objective) and discussing the relevance of it in a practical context.
* The idea of using embeddings, cyclical acceptance, and predicate-conditioned learning builds directly on recent work [1] [2] [3] and I think these principles are helpful in the aim to scale automata-driven RL further to large scale applications.

[1] Compositional Automata Embeddings for Goal-Conditioned Reinforcement Learning. Yalcinkaya et. al 2024.

[2] LTL-Constrained Policy Optimization with Cycle Experience Replay. Shah et. al 2024.

[3] Instructing Goal-Conditioned Reinforcement Learning Agents with Temporal Logic Objectives. Qiu et. al. 2023

**Weaknesses:**

Although building on very recent work is a good way to step the field forward, it does also beg a bit the question of significance. This work bears strong similarities to [3], with the primary change being to condition over reach-avoid sequences rather than individual atomic propositions or predicates that represent transitions within an automaton. The latter approach, which is what is done in [3], requires a planning-based approach each time a new automaton is seen. The authors do compare against [3] experimentally, and show that on individual challenging tasks their approach is better, which is appreciated. However, I'd like to see a more thorough experimental analysis of the DeepLTL approach itself. Since the DeepLTL approach is quite similar to prior work, this analysis-style work would greatly benefit the field. At what level of complexity of specification does the approach break down? Does a larger alphabet (and therefore a larger class of reach-avoid sequences) make the problem harder by expanding the space of possible embeddings?

Regarding the writing: I don't think including the discussion on eventual discounting [4] (problem 3.1 and theorem 3.1) is totally necessary and the small extension of the theory that the authors provide is more or less orthogonal to their main contribution, which obscures the writing a bit. The authors use a discounted version of LTL as their objective but do not cite recent work that thoroughly explores this problem setting [5]. In section 4.1, the authors discuss reasoning over pre-computed accepting cycles, which bears strong similarities to an identical approach in [2]. Although [2] is cited it would be good for the authors to mention it in section 4.1 given these similarities.

Lastly, the approach from [1] is a highly similar approach to automata-goal-conditioned RL that also uses an embedding based approach. Although this work is contemporaneous, a previous version did appear [6] earlier and I think some sort of comparison, if not an explicitly direct one, would be important in strengthening this work.

[1] Compositional Automata Embeddings for Goal-Conditioned Reinforcement Learning. Yalcinkaya et. al 2024.

[2] LTL-Constrained Policy Optimization with Cycle Experience Replay. Shah et. al 2024.

[3] Instructing Goal-Conditioned Reinforcement Learning Agents with Temporal Logic Objectives. Qiu et. al. 2023.

[4] Eventual Discounting Temporal Logic Counterfactual Experience Replay. Voloshin et. al. 2023.

[5] Policy Synthesis and Reinforcement Learning for Discounted LTL. Alur et. al. 2023.

[6] Automata Conditioned Reinforcement Learning with Experience Replay. Yalcinkaya et. al. 2023.

**Questions:**

* Can the authors compare against [1]/[6] in the previous section(s) and reason about why their approach may be preferable? The approaches are different in how they condition and compute embeddings but an argument by the authors advocating their own approach is important given the similarity of the work.
* The authors include a curriculum-based ablation in the appendix that supports the presence of a curriculum. What other choices of curricula were considered? Do the authors have ideas on how a choice of curriculum would affect learning?
* Section D.3 in the appendix seems to be missing. Can the authors provide this?
* At what level of complexity of specification does the deepLTL approach break down? Does a larger alphabet (and therefore a larger class of reach-avoid sequences) make the goal-conditioned RL problem harder by expanding the space of possible embeddings?

---

> ### Author Response · Authors · 2024-11-23
> **Response (1/2)**
>
> Thank you for taking the time to review our paper! We are pleased to read your positive comments, and very much appreciate the detailed feedback. We agree with the proposed changes and have thus **updated the paper to address the feedback, including further experimental results and a discussion and experimental comparison to [1]**. Below we provide a detailed response to the points and questions raised (in two parts).
>
> **Further experimental analysis of DeepLTL**
>
> > Does a larger alphabet (and therefore a larger class of reach-avoid sequences) make the problem harder by expanding the space of possible embeddings?
>
> We investigate this question in the updated Appendix G.5. We conduct experiments in a modified version of the *FlatWorld* environment with a varying number of atomic propositions. We chose *FlatWorld* as it can be readily augmented with the number of propositions as a parameter, and since its state space does not depend on the number of propositions. This allows us to use the same model architecture in each case, and control for differences arising solely from different state spaces.
>
> The results confirm our expectation that policy performance generally decreases as the number of propositions increases, since additional propositions significantly increase the task space, making the goal-conditioned RL problem more difficult. However, we note that the performance decrease of DeepLTL is generally similar or less than the performance reduction of the baseline GCRL-LTL. Integrating more advanced goal-conditioned RL algorithms, such as counterfactual experience [4], would likely further improve performance in environments with a large number of propositions; however, this is not the main focus of our paper and we thus leave it for future work.
>
> > At what level of complexity of specification does the approach break down?
>
> We provide a discussion and experimental analysis in the updated Appendix G.6. An advantage of our task representation based on reach-avoid sequences is that it makes the ways of satisfying a given specification explicit, allowing our method in principle to scale to large and complex formulae. We illustrate this with a concrete example formula that results in an LDBA with 656 states, with DeepLTL still achieving a success rate of 98%.
>
> However, generally the complexity of satisfying a given specification primarily depends on the underlying MDP. For example, if proposition *p* is difficult to achieve in the MDP, then even the simple formula *F p* is challenging. Finally, we note that we assume that we can construct the LDBA, which is doubly-exponential in the worst case (Sickert et al. 2016). While many other methods also rely on this assumption, clearly there are cases in which it is infeasible to construct the LDBA.
>
> **Comparison to [1]**
>
> Many thanks for providing this reference. We now mention it in the updated related work section and give a more detailed comparison in the extended related work (Appendix D). Furthermore, we conduct an experimental comparison of our approach and [1] in Appendix G.7.
>
> In contrast to our approach, [1] is based on DFAs, which are limited to finite-horizon tasks and thus strictly less expressive than LDBAs. Their approach to computing embeddings is also different: [1] computes an embedding for the entire automaton, whereas we compute embeddings of reach-avoid sequences extracted from the LDBA. This separates high-level reasoning (how to satisfy the specification) from low-level reasoning (how to act in the MDP) and allows the goal-conditioned policy to focus on achieving one particular sequence of propositions.
>
> Our experimental results demonstrate that [1] and our method achieve similar success rates on finite-horizon tasks, but our method generally requires significantly fewer steps until completion. We also note that DeepLTL achieves much higher success rates on tasks with a large associated automaton, since our policy is conditioned on only a single satisfying sequence rather than the whole automaton structure. These results highlight the advantages of our embeddings based on reach-avoid sequences.
>
> We continue our response below.

---

> ### Author Response · Authors · 2024-11-23
> **Response (2/2)**
>
> **Writing**
>
> > I don't think including the discussion on eventual discounting [4] (problem 3.1 and theorem 3.1) is totally necessary [...] obscures the writing a bit.
>
> We include the discussion on eventual discounting since we believe it is important to formally develop the problem statement and motivate the approximate objective in Problem 2. This is especially important since one of the advantages of our method is that it is non-myopic, which is not a concern under the eventual discounting setting. Do you have any suggestions how we could make this discussion clearer?
>
> > The authors use a discounted version of LTL as their objective but do not cite recent work that thoroughly explores this problem setting [5].
>
> Many thanks for the reference! We now cite [5] in Section 3 and added a discussion of discounted LTL to the appendix (Appendix C).
>
> > In section 4.1, the authors discuss reasoning over pre-computed accepting cycles, which bears strong similarities to an identical approach in [2].
>
> We appreciate that there are similarities between our approach and [2], and followed your suggestion of mentioning these explicitly in Section 4.1 in the updated paper. However, we also note that there are significant differences between our approach and [2] : [2] makes use of accepting cycles in the setting of a single, fixed task for the purpose of reward shaping. In contrast, we use the set of paths to accepting cycles from the current LDBA state as a representation to condition a goal-conditioned policy in a multi-task setting, and show that we can use them for learning useful goal embeddings. We are not aware of any prior work that uses accepting cycles in a similar way.
>
> **Questions**
>
> > The authors include a curriculum-based ablation in the appendix that supports the presence of a curriculum. What other choices of curricula were considered? Do the authors have ideas on how a choice of curriculum would affect learning?
>
> The curricula are designed to gradually expose the policy to more challenging tasks. As such, the curricula we consider generally start with short reach-avoid sequences and move on to longer and more complex sequences as the policy improves. Intuitively, it does not make sense to train on a sequence $(a,b)$ if the policy cannot yet satisfy $(a)$ alone. This explains why we observe in our ablation study that using a curriculum speeds up learning; we believe that other curricula with similar properties should yield comparable results. In particular, we would be excited to explore techniques such as automated curriculum design in future work.
>
> > Section D.3 in the appendix seems to be missing. Can the authors provide this?
>
> Many thanks for catching this, we include the hyperparameters in the updated Appendix F.3.
>
>
> Thank you again for the detailed feedback! We hope that the updates to the paper, additional experimental results, and our comments are helpful. Please let us know if these have addressed your concerns. We are more than happy to engage in further discussion!

---

> > ### Comment · Reviewer_Xrc5 · 2024-11-26
> > **Thanks to the authors re: response**
> >
> > Thanks to the authors for the detailed and thorough response.
> >
> > > The results confirm our expectation that policy performance generally decreases as the number of propositions increases, since additional propositions significantly increase the task space, making the goal-conditioned RL problem more difficult.
> >
> > This intuition makes sense to me - it would be good to include a brief discussion in the paper on this, and potentially discuss how it would impact the embedding space.
> >
> > > Their approach to computing embeddings is also different: [1] computes an embedding for the entire automaton, whereas we compute embeddings of reach-avoid sequences extracted from the LDBA. This separates high-level reasoning (how to satisfy the specification) from low-level reasoning (how to act in the MDP) and allows the goal-conditioned policy to focus on achieving one particular sequence of propositions.
> >
> > Thanks for including this discussion. I'd again encourage the authors to include this in the appendix.
> >
> > >We include the discussion on eventual discounting since we believe it is important to formally develop the problem statement and motivate the approximate objective in Problem 2. This is especially important since one of the advantages of our method is that it is non-myopic, which is not a concern under the eventual discounting setting. Do you have any suggestions how we could make this discussion clearer?
> >
> > I don't think developing the approximate objective in problem 2 relies on the discussion on eventual discounting - if anything, it's just context that could very easily be boiled down to a few sentences discussing how LTL is typically satisfied, and then advocating for why a discounted setting would be useful (you can refer to the body of work that does consider discounted LTL such as the reference I provide in my original review.) I'd encourage the authors to summarize the eventual discounting / typical approach for LTL in a short paragraph and then introduce their problem. This will give more space for valuable experimental discussion.
> >
> > Overall, I am satisfied with the response from the authors. I will update my score to recommend acceptance for the work and I encourage the authors to include the discussion from the rebuttals in the revised version of the paper.

---

> > > ### Author Response · Authors · 2024-11-26
> > > **Thanks**
> > >
> > > We would like to thank the reviewer for reassessing our paper, and are glad that our response has been satisfactory. We agree with the comments and are working on revising the paper accordingly, in particular Section 3. Many thanks again for your feedback, which we feel has substantially improved the paper.

---

### Official Review · Reviewer_aeBd · 2024-11-05

**Soundness:** 3
**Presentation:** 3
**Contribution:** 3
**Rating:** 8
**Confidence:** 5

**Summary:**

This paper proposes a method that leverages linear temporal logic (LTL) to formulate reinforcement learning (RL) tasks. The authors claim that their method is applicable to infinite-horizon tasks and are non-myopic. The preliminaries and problem setting are presented in a clear and logical flow, and the experimental results are well-reported. However, the authors seem to have completely missed highly relevant literature in this area (see references below).

**Strengths:**

1) The paper presents an interesting approach to learn policies to satisfy omega-regular specifications based on visiting accept states in an automaton without discounting states between the visits.
2) It incorporates policies parameterized as neural networks.
3) It uses the structure of the automaton specification.

**Weaknesses:**

The main weakness of this paper is that it ignores significant body of literature that deals with training policies for omega-regular objectives. Without a detailed comparison, it is difficult to evaluate the novelty in this paper. In fact, the technique of discounting seems quite similar to the zeta parameter used in the Hahn et al. paper from TACAS 2019. The authors should clarify how their approach is different.

References:
1. Hahn, E. M., Perez, M., Schewe, S., Somenzi, F., Trivedi, A., & Wojtczak, D. (2019, April). Omega-regular objectives in model-free reinforcement learning. In International conference on tools and algorithms for the construction and analysis of systems (pp. 395-412).
2. Hahn, E. M., Perez, M., Schewe, S., Somenzi, F., Trivedi, A., & Wojtczak, D. (2020). Faithful and effective reward schemes for model-free reinforcement learning of omega-regular objectives. In Automated Technology for Verification and Analysis: 18th International Symposium, ATVA 2020, Hanoi, Vietnam, October 19–23, 202
3. Le, Xuan-Bach, Dominik Wagner, Leon Witzman, Alexander Rabinovich, and Luke Ong. "Reinforcement Learning with LTL and $\omega $-Regular Objectives via Optimality-Preserving Translation to Average Rewards." arXiv preprint arXiv:2410.12175 (2024).
4. Hahn, E. M., Perez, M., Schewe, S., Somenzi, F., Trivedi, A., & Wojtczak, D. (2021). Mungojerrie: Reinforcement learning of linear-time objectives. arXiv preprint arXiv:2106.09161.

**Questions:**

Questions:
1) In section 4.2 and 4.3, the explanation of the sequence module, which encodes reach-avoid sequence, is unclear. What are the inputs and the outputs of this module?  Could you provide an example to clarify?
2) Why did you use an RNN? Transformer-based NN architectures outperform RNNs in many problems.
3) In section 4.5, the statement “the value function is a lower bound of the discounted probability of reaching an accepting state k times via…” does not sound correct. How is the right hand side of the inequality equal to “the discounted probability of reaching an accepting state k times” ? Can you explain your reasoning?
4) GCRL-LTL also works for infinite-horizon tasks. The experiment results imply that your method outperforms GCRL-LTL. Is there a theoretical explanation for why your method is better than GCRL-LTL?
5) It is difficult to evaluate the novelty of this paper without a thorough comparison to approaches such as those used in the tool Mungojerrie [4]. Will such a comparison be possible in a short time?

(See further questions in the post-rebuttal review)

---

> ### Author Response · Authors · 2024-11-15
>
> Thank you for taking the time to review our paper! We provide answers to your questions below. In particular, we would like to point out some possible misunderstandings that might have negatively impacted the assessment.
>
> **Existing literature on $\omega$-regular objectives**
>
> We appreciate the given references and agree that they are relevant in the context of our work. However, please note that these approaches tackle a **different/simpler problem** than the one we consider and are **not applicable** to our setting. In particular, our approach is realised in a **multi-task RL** setting and we train a single policy that can **zero-shot execute arbitrary unseen LTL specifications at test time**. In contrast, the methods in [1-4] only learn a policy for a **single, fixed task**. This is a crucial difference: our approach trains a single policy once, which can then satisfy arbitrary tasks such as the ones in our evaluation (see Tables 2 and 3). The given references would have to train a separate new policy for every specification, and cannot generalise to new tasks at test time.
>
> We have submitted an updated version of the paper to hopefully make this distinction clearer. The updated version includes a changed title that explicitly mentions multi-task RL, and minor corresponding edits to the abstract and introduction. We have also updated the related work section (Section 6) to more explicitly point out the differences to methods that only handle a single task, and include an extended discussion of related work in the appendix (see Appendix C in the updated paper), which includes the provided references.
>
> In the context of multi-task RL with LTL specifications, which is the problem we consider in our paper, we are only aware of a single work that can handle $\omega$-regular specifications (Qiu et al. 2023). As we discuss in the paper (Section 4.6 and lines 518-525) our approach has various theoretical advantages (also see below), and we demonstrate that it performs better in practice (see Table 1, Figure 6, Table 5, Figure 9).
>
> **Answers to questions**
>
> > Q1
>
> As illustrated in Figure 4, the sequence module takes as input a reach-avoid sequence $\sigma$ and outputs a corresponding embedding $e_\sigma$ that is used to condition the policy. For example, if the current task is F (a & F b), the corresponding reach-avoid sequence is (({a}, {}), ({b}, {})) (where all avoid sets are empty). The sequence module maps this sequence to some embedding $e\in\mathbb R^n$ which conditions the trained policy to first reach proposition a and subsequently reach proposition b.
>
> > Q2
>
> We mainly opted for RNNs for the sake of simplicity: while Transformers excel at long-distance tasks, the sequences arising from the LTL formulae we consider are generally relatively short (2-15 tokens). Furthermore Transformers are known to be difficult to train and require large amounts of training data (Liu et al. 2020). We also note that our choice is consistent with previous works, which mainly use GRUs for sequence modelling (Kuo et al. 2020, Vaezipoor et al. 2021, Xu and Fekri 2024).
>
> [1] Liu et al (2020). ‘Understanding the Difficulty of Training Transformers’. In *EMNLP'20*.
>
> > Q3
>
> Let $\sigma$ be a truncated reach-avoid sequence that visits an accepting state $k$ times, and denote the length of $\sigma$ as $n$. As per our training procedure (Section 4.4) we have that $i = n + 1$ iff the agent has satisfied all assignments in $\sigma$, i.e. successfully finished "executing" the sequence. This means the agent has visited an accepting state $k$ times. The expected value
> $$
> \mathbb E_{\tau\sim\pi|\sigma}\left[ \sum_{t=0}^\infty \mathbb 1[i = n+ 1]\right]
> $$
> is thus exactly the probability of the policy reaching an accepting state $k$ times by following $\sigma$.
>
> > Q4
>
> Yes, in comparison to GCRL-LTL our method is **non-myopic** and considers **safety constraints during planning** (cf. Section 4.6 and lines 520-523). These theoretical advantages explain why our approach outperforms GCRL-LTL in terms of efficiency and satisfaction probability. In Appendix F.2 we also provide a further comparison to GCRL-LTL on tasks with safety constraints, which highlights the differences in the planning approaches.
>
> > Q5
>
> As discussed above, our paper proposes a novel method for zero-shot execution of arbitrary LTL specifications in a multi-task RL setting. This is fundamentally different from the approaches implemented in [4], which train a policy for a single, fixed specification and cannot generalise to different tasks. Do you still think such a comparison is useful and required, given the fundamentally different problem statements?
>
> We appreciate your comments and hope that our response along with the corresponding edits in the revised version of the paper has made our contribution clearer. If anything else is unclear, we would be more than happy to engage in further discussion. Please let us know if this mitigates your concerns!

---

> > ### Author Response · Authors · 2024-11-25
> > **Follow up**
> >
> > Since the discussion period is coming to an end, we wanted to check whether we have been able to address your concerns with our response and the edits to the paper. Please let us know if you have any additional questions, we are happy to engage in further discussion.

---

> > > ### Author Response · Authors · 2024-11-27
> > > **Less than 24 hours remaining**
> > >
> > > Since there are less than 24 hours remaining for us to make any edits to the paper, we wanted to follow up once more and kindly ask the reviewer if they have any additional comments/concerns.

---

> > > > ### Comment · Reviewer_aeBd · 2024-11-27
> > > > **Some concerns addressed**
> > > >
> > > > The authors rebuttal does address some concerns. An important aspect: training policies for arbitrary LTL formulas was not clearly highlighted in the initial submission, which led to concerns about lack of comparison with previous approaches. Clarifying this does make the approach more novel than previously thought. I will upgrade my score to reflect this.
> > > >
> > > > After further contemplation about the paper, I have the following suggestions/criticisms:
> > > > 1) The authors should provide a high-level comparison of the key ideas and techniques used in their approach versus Mungojerrie, even if a full empirical comparison is not feasible in the short term. Potential challenges in implementing such a comparison and a timeline for future work addressing this comparison would be welcome. There is value in still comparing against a baseline SOTA method that works for a given specification.
> > > > 2) The authors should discuss their rationale for choosing RNNs over transformers, including any empirical comparisons they may have conducted, and whether the sequential nature of the reach-avoid sequences influenced this decision.
> > > > 3) It is important to discuss specific technical similarities and differences, especially regarding the discounting technique and the
> > > > zeta parameter mentioned in the review.
> > > > 4) The authors could consider providing a diagram to illustrate the inputs, outputs, and internal processing steps of the sequence module, along with a concrete example of how a reach-avoid sequence is encoded and processed.
> > > > 5) Could the authors  provide a theoretical analysis of the key factors that contribute to their method's superior performance over GCRL-LTL, particularly for infinite-horizon tasks?
> > > > 6) The outcome of the RL policy seems subject to the dynamics of the agent in the MDP, which isn’t encoded in the product MDP. How do you guarantee a high success rate even if you can encode arbitrary specs in your formulation?

---

> ### Author Response · Authors · 2024-11-30
> **Response (1/2)**
>
> Thank you for your response. We are pleased that the updates to the paper have clarified our contribution. Below we respond to the suggestions and criticisms raised (in two parts).
>
> **RNNs vs Transformers**
>
> Thank you for your suggestion! As discussed in our original response, we opted for RNNs over Transformers since the reach-avoid sequences arising from common LTL tasks are generally relatively short (2-15 tokens). In this scenario, we believe that the simplicity of training RNNs outweighs the empirical performance gains observed in Transformers on long-distance tasks with vast amounts of data.
>
> We confirm this hypothesis by experimentally evaluating DeepLTL with a Transformer model instead of an RNN in the *ZoneEnv* environment. For this, we broadly follow the BERT architecture: we use a Transformer encoder to learn an embedding of the sequence $\sigma$ as the final representation of a special [CLS] token. We use a small model with an embedding size of 32, 4 attention heads, and a 512-dimensional MLP with dropout of 0.1. Note that even though we use such a small Transformer model, it still has more than 5x the number of parameters of the RNN.
>
> We report the achieved success rates in the table below:
>
> | | DeepLTL-Transformer | DeepLTL-GRU | |
> | --- | --- | --- | --- |
> | $\varphi_{6}$  | 0.78$_{\pm0.09 }$   | **0.92**$_{\pm0.06 }$ |     |
> | $\varphi_{7}$  | 0.46$_{\pm0.04 }$   | **0.91**$_{\pm0.03 }$ |     |
> | $\varphi_{8}$  | 0.93$_{\pm0.03 }$   | **0.96**$_{\pm0.04 }$ |     |
> | $\varphi_{9}$  | 0.74$_{\pm0.02 }$   | **0.90**$_{\pm0.03 }$ |     |
> | $\varphi_{10}$ | 0.80$_{\pm0.15 }$   | **0.91**$_{\pm0.02 }$ |     |
> | $\varphi_{11}$ | 0.90$_{\pm0.07 }$   | **0.98**$_{\pm0.01 }$ |     |
>
> The results confirm our intuition that RNNs perform better than Transformers in our short-sequence and relatively low-data setting (compared to e.g. foundation models). We are happy to include these experimental results as an ablation study in the final version of the paper.
>
> **Comparison to Mungojerrie**
>
> Our approach differs in several key respects from the tool Mungojerrie. First and foremost, we address the problem of learning a policy that can zero-shot execute arbitrary LTL specifications, whereas Mungojerrie only learns a policy for a single, fixed LTL specification. As such, our techniques are fundamentally different: we first train a general sequence-conditioned policy on a variety of reach-avoid sequences with a focus on generalisation to arbitrary sequences. At test time, we are given an unseen LTL formula, construct the corresponding LDBA, extract possible reach-avoid sequences, select the best sequence according to the learned value function, and finally leverage the trained sequence-conditioned policy to satisfy the formula (see Figure 3).
>
> In contrast, Mungojerrie only has to deal with a single LTL specification. It constructs the LDBA for this, and directly trains a policy in the product MDP of the original MDP and the LDBA. Mungojerrie introduces a variety of different reward schemes for the reinforcement learning objective that ensure the resulting policy is probability-optimal. In contrast, we trade off optimality and efficiency of the resulting policy (see the discussion in Section 3 and Appendix B).
>
> > Empirical comparison
>
> We agree that there is value in comparing to methods that only work for a single specification. However, please note that the primary purpose of our method is to learn a policy that is able to generalise to arbitrary formulae at test time, and this is what we test in our experimental evaluation: we train a single policy and test it on a range of complex tasks. We thus compare to methods that tackle the same problem in our experiments, as it is not clear how to fairly compare to methods that can only handle a single specification. We also note that Mungojerrie cannot be easily applied to the experiments that we consider, since it can only handle models with finite state and action spaces specified in PRISM, whereas we consider arbitrary MDPs with potentially continuous state and action-spaces (e.g. ZoneEnv, FlatWorld).
>
> > Technical details
>
> We hope our discussion above clarifies the technical differences between our method and Mungojerrie. The $\zeta$ parameter in Mungojerrie is used to ensure that the reward-optimal policy is also probability-optimal. This is comparable to eventual discounting (Problem 1) (Voloshin et al. 2023), which ensures probability-optimality by only discounting visits to accepting states, without introducing an additional hyperparameter such as $\zeta$. Note that we first extend eventual discounting to the multi-task setting (Problem 1 and Theorem 1), but then consider a modified version which trades of probability-optimality with efficiency for the rest of the paper (Problem 2).
>
> We continue our response below.

---

> > ### Author Response · Authors · 2024-11-30
> > **Response (2/2)**
> >
> > **Comments**
> >
> > > The authors could consider providing a diagram to illustrate the inputs, outputs, and internal processing steps of the sequence module, along with a concrete example of how a reach-avoid sequence is encoded and processed.
> >
> > Please see the illustration of the sequence module in Figure 4. As an example, if the current task is F (a & F b), the corresponding reach-avoid sequence is (({a}, {}), ({b}, {})) (where all avoid sets are empty). The sequence module maps this sequence to some embedding which conditions the trained policy to first reach proposition a and subsequently reach proposition b. We are more than happy to provide further examples if anything else is unclear!
> >
> > > Could the authors provide a theoretical analysis of the key factors that contribute to their method's superior performance over GCRL-LTL, particularly for infinite-horizon tasks?
> >
> > As we mentioned in our original response, in comparison to GCRL-LTL our method is **non-myopic** and considers **safety constraints during planning** (cf. Section 4.6 and lines 522-525). By being non-myopic, our approach takes the whole specification into account, whereas GCRL-LTL only focuses on completing the next subtask. These theoretical advantages explain why our approach outperforms GCRL-LTL in terms of efficiency and satisfaction probability. Please let us know if anything else is unclear regarding the advantages of our method over GCRL-LTL.
> >
> > > The outcome of the RL policy seems subject to the dynamics of the agent in the MDP, which isn’t encoded in the product MDP.
> >
> > Please note that the product MDP includes the **entire original MDP** and thus in particular the dynamics of the agent (since these are defined in the underlying MDP). Please let us know if we misunderstood your comment.
> >
> > > How do you guarantee a high success rate even if you can encode arbitrary specs in your formulation?
> >
> > This is a very good question! Our approach manages to achieve high success rates by (1) decomposing specifications into reach-avoid sequences, which provide an explicit representation of ways of satisfying the specification and then (2) leveraging the generalisation abilities of a policy trained on arbitrary reach-avoid sequences. We furthermore incorporate a planning step, which exploits the trained value function to ensure we select the reach-avoid sequence that is most likely to be able to be satisfied by the policy (see Section 4.5).
> >
> > Thank you again for the feedback! We hope our additional experiments and comments have addressed your concerns. Please let us know if you have any further questions, we are more than happy to engage further until the end of the discussion period.

---

> > > ### Comment · Reviewer_aeBd · 2024-12-02
> > > **Further suggestions/questions**
> > >
> > > 1) In your method, you compute lassos in the automaton and select a lasso to try to force that has the highest learned probability of succeeding. This seems a bit difficult to do for stochastic environments where one doesn't know which lasso will occur. Could you clarify?
> > > 2) The claim of being the first non-myopic method is a bit of overselling, because the paper compares with a specific prior method, while bucketing other prior methods that consider a fixed specification (these other prior methods are also non-myopic!).
> > >
> > > Other responses do clarify my questions, so I have raised the score further.

---

> > > > ### Author Response · Authors · 2024-12-04
> > > >
> > > > We are glad that our responses were able to clarify your questions; thank you for engaging and reassessing our paper!
> > > >
> > > > > In your method, you compute lassos in the automaton and select a lasso to try to force that has the highest learned probability of succeeding. This seems a bit difficult to do for stochastic environments where one doesn't know which lasso will occur. Could you clarify?
> > > >
> > > > Exactly, we use the learned value function $V^\pi$ to estimate which lasso the policy is most likely to be able to satisfy. We agree that this will generally be more difficult in stochastic environments. However, in principle stochasticity is not a problem since the value function is trained to predict the *expected value* of success; if there is a large amount of variance for a specific lasso we would expect this to be reflected in the mean value predicted by $V^\pi$. We also note that our method dynamically recomputes the best sequence $\sigma^*$ based on the current LDBA state. In particular, this means if the agent tries to follow a particular lasso, but then reaches a different LDBA state where that lasso is no longer valid, it will instead aim to follow a different lasso that leads to satisfying the formula. However, our approach does rely on learning a relatively accurate estimate $V^\pi$ of the value function.
> > > >
> > > > > The claim of being the first non-myopic method is a bit of overselling, because the paper compares with a specific prior method, while bucketing other prior methods that consider a fixed specification (these other prior methods are also non-myopic!).
> > > >
> > > > Thanks for pointing this out! We claim that DeepLTL is the first non-myopic method that can handle $\omega$-regular specifications in the multi-task RL setting. We will revise the final version of the paper to more clearly state that prior non-myopic methods for $\omega$-regular specifications exist that handle a fixed specification.
> > > >
> > > > Many thanks again for your valuable comments and feedback!

---

### Meta-Review · Area_Chair_SbXL · 2024-12-20

**Metareview:**

This paper presents DeepLTL, a method to perform multi-task RL with LTL specifications. The technique leverages two recent innovations, eventual-discounting and goal-conditioned RL, to create RL agents that can zero-shot generalize to wide range of specifications. The paper demonstrates that the technique provides competitive results in discrete and continuous environments with finite and infinite horizon specifications.

**Additional Comments On Reviewer Discussion:**

Most reviewers raised concerns about the discussion of related work, and missing citations of relevant papers. The authors expanded their discussion of related work, and clarified their problem setting: training RL agents in the multi-task setting with the ability to zero-shot generalize to a variety of specifications. The discussion, and subsequent updates to the paper, have greatly improved its quality.

---

### Decision · Program_Chairs · 2025-01-22

Accept (Oral)